# Positive Interventional Effect of Engineered Butyrate-Producing Bacteria on Metabolic Disorders and Intestinal Flora Disruption in Obese Mice

Lina Wang,[a,b] Xiaoming Cheng,[a,b] Liang Bai,[a,b,c] Mengxue Gao,[a,b] Guangbo Kang,[a,b,d] ⓘ Xiaocang Cao,[e] ⓘ He Huang[a,b]

aDepartment of Biochemical Engineering, School of Chemical Engineering and Technology, Tianjin University, Tianjin, China
bFrontiers Science Center for Synthetic Biology and Key Laboratory of Systems Bioengineering (Ministry of Education), Tianjin University, Tianjin, China
cTianjin Modern Innovative, TCM Technology Co. Ltd., Tianjin, China
dInstitute of Shaoxing, Tianjin University, Zhejiang, China
eDepartment of Gastroenterology and Hepatology, Tianjin Medical University General Hospital, Tianjin Medical University, Tianjin, China

Lina Wang, Xiaoming Cheng, and Liang Bai equally contributed to this article. Author order was determined by the corresponding author He Huang after inner negotiation.

**ABSTRACT** The substantially increased prevalence of obesity and obesity-related diseases has generated considerable concern. Currently, synthetic biological strategies have played an essential role in preventing and treating chronic diseases such as obesity. A growing number of symbiotic bacteria used as vectors for genetic engineering have been applied to create living therapeutics. In this study, using *Bacillus subtilis* as a cellular chassis, we constructed the engineered butyrate-producing strain BsS-RS06551 with a butyrate yield of 1.5 g/liter. A mouse model of obesity induced by a high-fat diet (HFD) was established to study the long-term intervention effects of this butyrate-producing bacteria on obesity. Combined with phenotypic assay results, we found that BsS-RS06551 could effectively retard body weight gain induced by a high-fat diet and visceral fat accumulation of mice, whereas it could improve glucose tolerance and insulin tolerance, reducing liver damage. We explored the BsS-RS06551 mechanism of action on host function and changes in intestinal flora by integrating multiple omics profiling, including untargeted metabolomics and metagenomics. The results showed that 24 major differential metabolites were involved in the metabolic regulation of BsS-RS06551 to prevent obesity in mice, including bile acid metabolism, branch chain amino acids, aromatic amino acids, and other metabolic pathways. Continuous ingestion of BsS-RS06551 could regulate gut microbiota composition and structure and enhance intestinal flora metabolic function abundance, which was closely related to host interactions. Our results demonstrated that engineered butyrate-producing bacteria had potential as an effective strategy to prevent obesity.

**IMPORTANCE** Obesity is a chronic metabolic disease with an imbalance between energy intake and energy expenditure, and obesity-related metabolic diseases have become increasingly common. There is an urgent need to develop effective interventions for the prevention and treatment of obesity. This study showed that long-term consumption of BsS-RS06551 had a significant inhibitory effect on obesity induced by a high-fat diet and was more potent in inhibiting obesity than prebiotic inulin. In addition, this study showed a beneficial effect on host glucose, lipid metabolism, and gut microbe composition. Considering its colonization potential, this engineered bacteria provided a new strategy for the effective and convenient treatment of obesity in the long term.

**KEYWORDS** butyrate, engineered bacteria, gut microbes, metabolomics, metagenomics, obesity, synthetic biology

Address correspondence to Xiaocang Cao, doccaoxc@163.com, or He Huang, huang@tju.edu.cn.

The authors declare no conflict of interest.

Obesity has emerged as a major health concern worldwide and become a global epidemic. It is taking a major toll on human health, increasing the risk of type 2 diabetes, osteoarthritis, anxiety, and cancers (1–4). Obesity is the result of complex interactions between environmental and genetic factors, which mainly include poor living habits, heredity, living environment, and social intercourse. Simultaneously, obesity is associated with an abnormal intestinal microbial composition and metabolic disorders and is generally followed by insulin resistance and dyslipidemia. However, general lifestyle and behavioral changes have limited effectiveness in reducing calorie intake and increasing energy expenditure (5, 6). Although some chemical drugs on the market (e.g., phentermine and orlistat) have specific therapeutic effects on obesity in the short term, there are apparent side effects when these drugs are used in the long term (7). Thus, there is a growing need for safe and efficient alternatives for managing obesity.

Live biotherapeutic products (LBPs) (8, 9) in the form of probiotics recently have been observed to be excellent for improving intestinal metabolism (10, 11). The persistence of live natural medicines would be a better treatment for chronic diseases such as obesity. Disorders of metabolism and imbalance in gut microbes are significant features of obesity (12–14). Short-chain fatty acids (SCFAs), produced by our gut microbes, have shown positive effects on preventing and counteracting obesity in the mechanism of energy homeostasis, lipid metabolism, and insulin sensitivity (15–17). Additionally, SCFAs play a vital role in remodeling the composition and function of gut microbes (18–20). Specifically, butyrate has been reported to decrease the pH of the intestine, provide energy to colonic epithelial cells, and regulate energy absorption and immune response. Moreover, butyrate has a good effect on preventing and suppressing obesity by improving the glucose homeostasis, insulin sensitivity, and function of the intestinal microbiota (13, 14, 19, 21). Butyrate supplementation seems to be an effective obesity treatment strategy, but it often has low bioavailability, and direct butyrate supplementation has a poor sustained effect (21, 22). *Bacillus subtilis* is a food-grade probiotic with a clear genetic background and a good load expression system (23). However, natural *B. subtilis* has low butyric acid production ability.

In the previous study, we used synthetic biological strategies to construct a butyrate-producing bacteria, recombinant *B. subtilis* BsS-RS06550 with high butyric acid production, and we found that it showed promising therapeutic effects in mice with obesity induced by high-fat diets (24). At the same time, we found that the safety and effectiveness of the engineered strain BsS-RS06550 was better than the original strain in short-term treatment. However, further research is required to investigate whether the engineered bacteria have the ability to regulate the gut microbes and host metabolism in order to prevent the development of obesity in the long term. Thus, in this study, we constructed a new recombinant strain *B. subtilis* BsS-RS06551 with higher butyric acid production compared to BsS-RS06550. Due to the metabolic benefits of inulin on improving fat metabolism and blood sugar levels in obesity (25), we choose it as a control. We evaluated the effectiveness of BsS-RS06551 and inulin supplement in mice in a long-term diet-induced obesity setting. Metagenomic sequencing and untargeted metabolomics were used to explore the possible mechanism underlying intervention strategies with regard to gut microbes and metabolites in mice. According to the results, the engineered bacteria could reduce weight gain caused by high-fat diets, improve glucose tolerance and insulin tolerance, alter the composition and function of the intestinal flora, and affect host metabolism. The effect can be equal to or surpass inulin in phenotype. Thus, we comprehensively put forward the hypothesis that supplementation with engineered BsS-RS06551 in obese mice could act on the microbiota-gut-brain communication to regulate obesity-related disturbances in glucose homeostasis and insulin resistance.

## RESULTS

**Genetic modification of engineered *B. subtilis* SCK6 to enhance butyrate production.** We genetically modified *B. subtilis* SCK6 and constructed the engineered strain BsS-RS06551 to enhance its butyric acid (BA) production. The *skfA* and *sdpC* genes encode bacteriophage killing factor and signal peptide proteins, respectively, both of which are released during bacteriophage formation and can promote autolysis of the bacterium and reduce biomass. Therefore,

*skfA* and *sdpC* gene knockout could reduce bacteriophage autolysis and increase biomass, thus promoting BA production (26). Acetic acid is a product of *B. subtilis* metabolism and competes with BA for the substrate acetyl-CoA, which weakens the metabolic flux of BA synthesis. The gene *acdA* encodes an acetyl-CoA ligase that regulates the production of acetic acid from acetyl-CoA, and knockout of this gene can block the production of acetic acid. In addition, knockout of the acetate kinase gene (*ackA*) has been shown to significantly reduce acetate production and simultaneously increase BA production (27). Taking these considerations into account, we knocked out four genes: *skfA*, *sdpc*, *acdA*, and *ackA*. In addition, we added a new butyric acid biosynthetic pathway to *B. subtilis* SCK6 by inserting the BCoAT-encoding gene into its genome after the gene encoding CRO (Fig. S1A). Compared with the control strain (SCK6), the engineered SCK6 strain (BsS-RS06551) exhibited an apparent increase in butyrate production, which reached $1.53 \pm 0.042$ g/liter during *in vitro* fermentation (Fig. S1B).

**Butyrate-producing engineered bacteria BsS-RS06551 suppressed the development of weight gain and insulin resistance in mice with high-fat diet-induced obesity.** To investigate the long-term effects of butyrate-producing bacteria on obesity, we provided three groups of mice with distilled water (the high-fat diet [HFD] group), prebiotic inulin (the HP group), and the engineered SCK6 strain BsS-RS06551 (the HEP group), respectively, for 14 weeks. In our previous study, we confirmed that high-fat diets induced glucose intolerance and lipid accumulation in mice (24). First, we observed that the engineered bacteria affected weight gain in mice. The data showed that during the first 5 weeks, there was no significant differences in body weight among three groups. After 5 weeks, intervention with BsS-RS06551 showed a significant inhibitory effect on the weight gain in mice on a high-fat diet ($P < 0.001$). It was also noted that continuous intake of inulin reduced the amount of weight gain (Fig. 1A). Compared with the group of HFD, the HP and HEP showed noticeable increases in food intake (Fig. 1B), which also showed that HEP ($P < 0.01$) had better control over obesity weight and food intake than HP ($P < 0.05$). The onset and development of obesity is often accompanied by insulin resistance. Next, we measured fasting plasma glucose and fasting plasma insulin levels in mice to assess the effect of engineered bacteria on obesity-associated insulin resistance. The levels of fasting blood glucose and insulin were all lower in HP and HEP groups than in the HFD group ($P < 0.01$) (Fig. 1C and D). Glucose tolerance and insulin tolerance of each group of mice were further examined and were significantly improved in the HEP group ($P < 0.001$) compared to the HFD group (Fig. 1E and F). The changes in insulin in the HP group ($P < 0.001$) were comparable to those in the HEP, but the changes in glucose ($P < 0.01$) were not as significant as in the HEP group. In addition, supplementation with BsS-RS06551 inhibited the increases in the homeostasis model assessment of insulin resistance (HOMA-IR) (Fig. 1G), indicating that BsS-RS06551 exerted an effect to ameliorate HFD-induced insulin resistance, similar to inulin ($P < 0.01$). The results of the serum biochemical indexes of the mice are shown in Fig. 1H to K, including total cholesterol (TC), triglyceride (TG), high-density lipoprotein (HDL), and low-density lipoprotein (LDL). Specifically, mice administered either BsS-RS06551 or inulin showed significantly lower levels in TG and LDL ($P < 0.001$) than HFD. TC and HDL levels were reduced in the HP group compared with HFD, but neither was as significant as in the HEP group ($P < 0.01$, $P < 0.05$). These results suggested that intervention with BsS-RS06551 decreased fasting blood sugar and insulin levels and improved glucose tolerance and insulin resistance in mice. In other words, the butyrate-producing engineered bacteria showed beneficial effects on the physiological and biochemical indicators of mice on a high-fat diet by effectively inhibiting weight gain and fat accumulation.

**Biochemical analysis of serum and hepatic tissue revealed the positive role of the engineered bacteria.** The liver is the most important metabolic organ in animals and humans. To evaluate the contribution of BsS-RS06551 toward the metabolic benefits in obesity treatment, we next evaluated the biochemical markers of liver function and histopathological changes in liver tissue. As shown in Fig. 2A, significantly lowered liver weights were observed in the HP and HEP groups compared with the HFD group ($P < 0.05$), but no significant difference was observed in the HP and HEP groups. The serum total bile acid (TBA) levels were significantly higher in the HEP group than in the HFD group ($P < 0.01$). Although there was no significant change in the level of the HP group, the level of TBA was higher relative to HFD (Fig. 2B). Alanine aminotransferase (ALT) and aspartate aminotransferase (AST) levels are

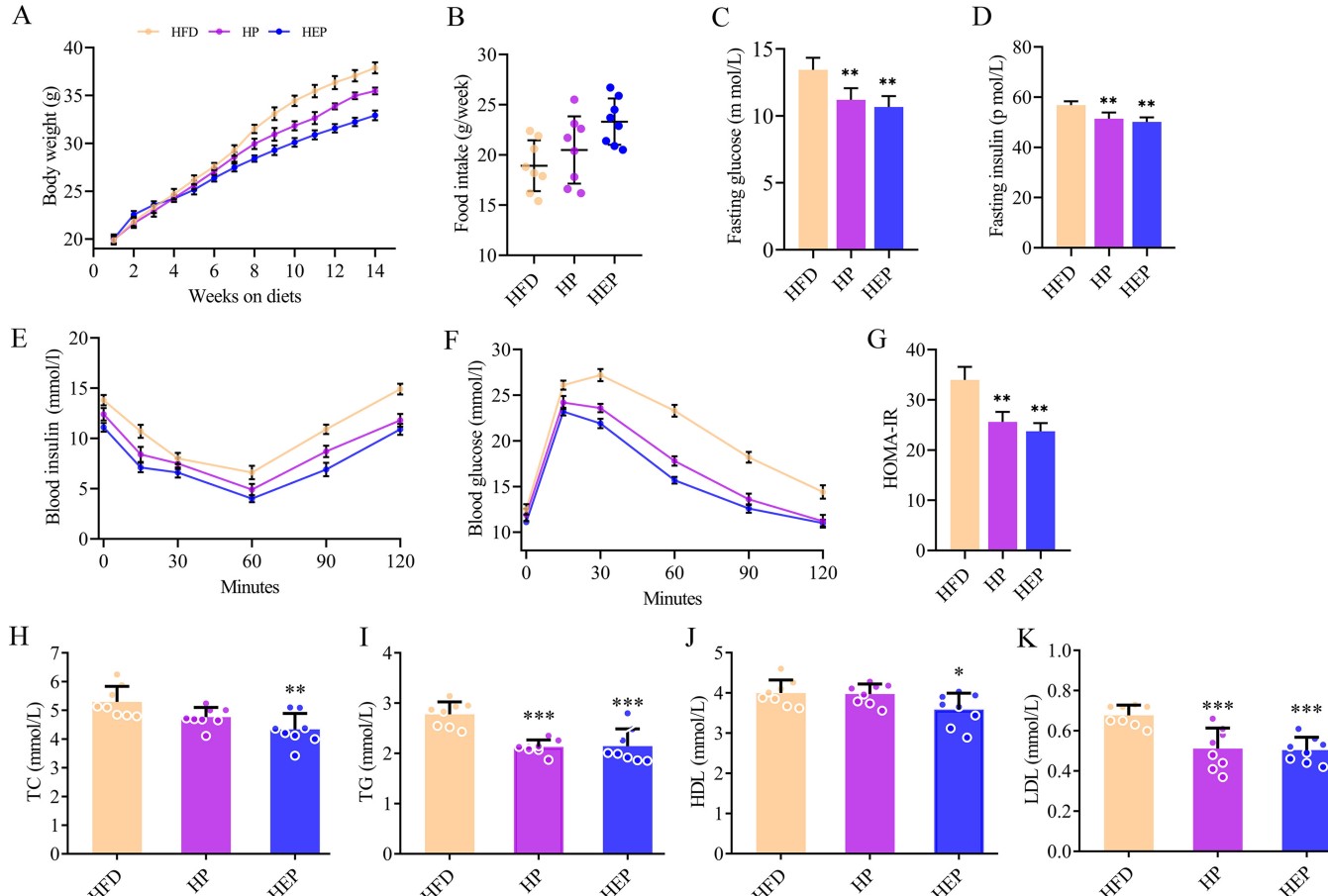

**FIG 1** The effect of BsS-RS06551 on blood physiology and lipid metabolism. (A) Weekly body weight. (B) Weekly food intake. (C) Fasting blood sugar. (D) Fasting insulin. (E) Insulin tests following intervention. (F) Glucose tolerance test following intervention. (G) Insulin resistance index (HOMA-IR). (H) TC levels. (I) TG levels. (J) HDL levels. (K) LDL levels. The data are represented as means ± SD, $n = 8$ mice/group for panels A to H. *, $P < 0.05$; **, $P < 0.01$; ***, $P < 0.001$. Yellow line, high-fat diet (HFD); purple line, HP; blue line, HEP. HFD, group treated with a high-fat diet and distilled water; HEP, group treated with a high-fat diet and engineered BsS-RS06551; HP, group treated with a high-fat diet and inulin; HOMA-IR, homeostasis model assessment of insulin resistance; TC, total cholesterol; TG, triglyceride; HDL, high-density lipoprotein; LDL, low-density lipoprotein.

significant indicators of liver injury. Compared with the HFD group, the HP and HEP groups showed drastic decreases in the ALT level, by approximately 50% ($P < 0.001$) (Fig. 2C). Although there was no relatively significant change in AST levels between the three groups, both HEP and HP were higher than HFD (Fig. 2D). Hematoxylin and eosin (H&E) and Oil Red O staining were performed to detect histopathological changes (Fig. 2E). The obvious red results indicated excess fat accumulation in the liver of mice fed high-fat diets, whereas supplementation with inulin or BsS-RS06551 attenuated hepatic steatosis and fat accumulation. Correspondingly, a higher number of lipid droplet vacuoles were observed more in the livers of HFD mice compared with those in HP or HEP mice. This indicated that long-term intake of BsS-RS06551 could alleviate liver dysfunction caused by high-fat diets, including reduction of fat accumulation and liver injury.

**Metagenomic sequencing revealed alterations of gut microbiota composition.** We performed metagenomic sequencing of fecal samples from mice to further explore the potential interaction effect between the preventive effect of engineered bacteria on obesity and the intestinal flora. We used different $\alpha$-diversity indexes to assess the changes in intestinal flora richness (Chao and ACE) and diversity (Shannon and Simpson) influenced by the engineered bacteria. As shown in Fig. S2A, the $\alpha$-diversity index showed increases according to the bacterial community profiles at the phylum level in HEP compared to HFD. However, both the Chao and ACE indexes showed different patterns at the genus and species levels (Fig. S2B and C). According to the sequenced species annotation information, we obtained the relative abundance of samples from each group at the taxonomic

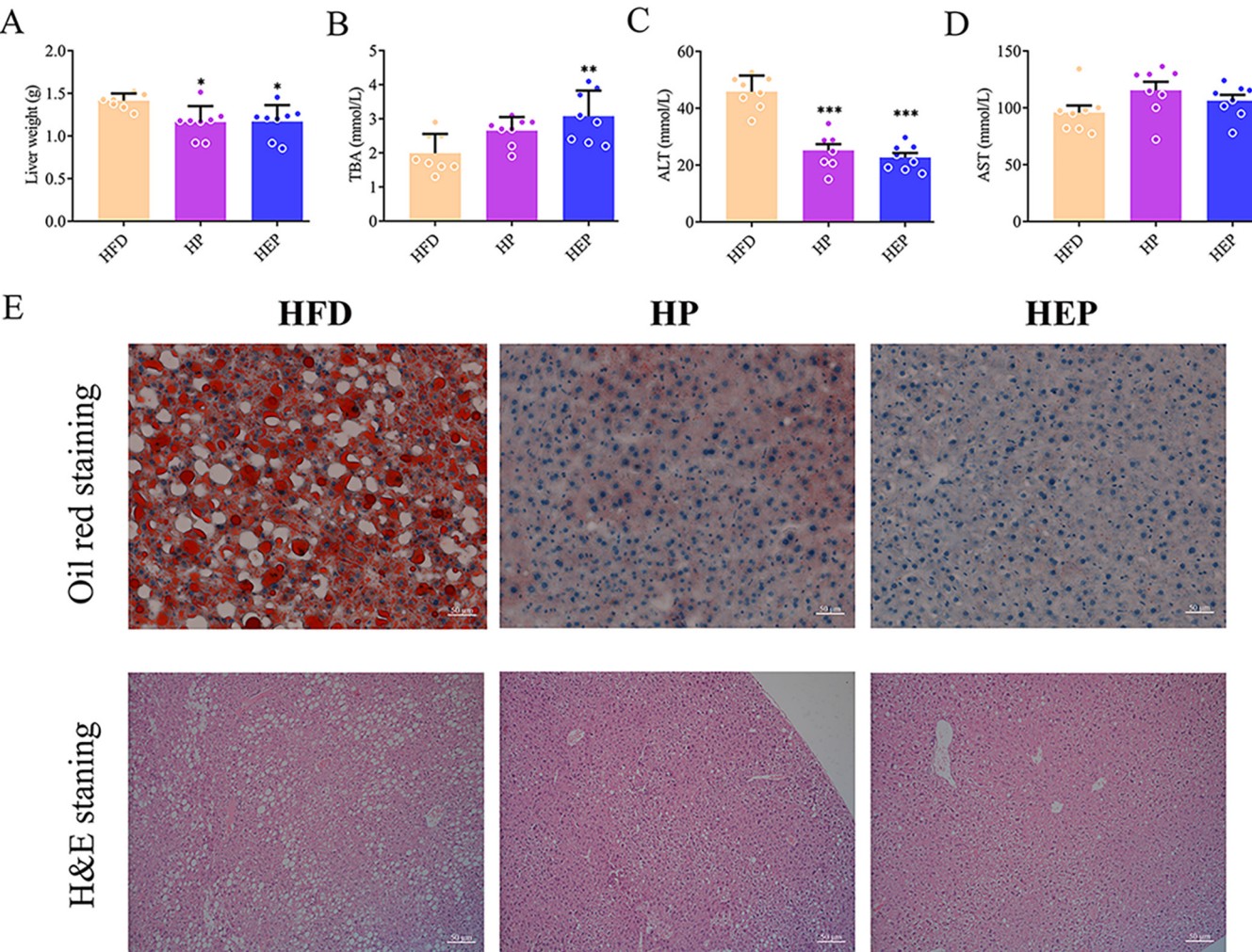

**FIG 2** Effects of BsS-RS06551 on HFD-induced hepatic steatosis. (A) Liver weight. (B) Serum total bile acid (TBA) level. (C) Serum alanine aminotransferase (ALT) levels. (D) Serum aspartate aminotransferase (AST) levels. (E) Hematoxylin and eosin (H&E) staining and Oil Red O staining of livers (×200). The data are represented as means ± SD, and $n = 8$ mice/group for panels A to C. *, $P < 0.05$; **, $P < 0.01$; ***, $P < 0.001$.

levels of phylum, genus and species. A principal coordinates analysis (PCoA) was performed to explore the similarity of the microbial communities in different samples based on Bray-Curtis dissimilarity (statistical method used to calculate beta diversity in R) (Fig. S2D). The analysis indicated that differences in the microbiota composition existed among the three groups.

We selected the top 10 species in abundance at each taxonomic level to draw histograms and the top 35 species in abundance to cluster and draw the heat map (Fig. 3). At the phylum level, Bacteroidetes, Firmicutes, Chlamydiae, Deferribacteres, and Proteobacteria were the dominant phylum in all grouped samples. Several types of bacteria related to obesity (28) (e.g., Bacteroidetes, Deferribacteres, Chlamydiae, and Prevotella) obviously changed the relative abundance, and the composition of flora underwent extensive changes. Compared with the HFD group, the abundance of Chlamydiae and Firmicutes in the HEP group was significantly reduced, and Bacteroidetes was significantly increased ($P < 0.001$). Notably, the Chlamydiae relative abundance in the HP seemed to be increasing, and Bacteroidetes was significantly increased ($P < 0.001$), which were opposite of the engineered bacteria prophylaxis group. Among them, Chlamydiae was an obligate intracellular parasitic microorganism, an important protozoan symbiote and pathogen of human body, which can promote the differentiation of adipose stem cells, lead to excessive accumulation of fat in inflammatory sites, and cause overweight or obesity (29). We could observe the obvious decreases in both HP and HEP. Bacteroides was correlated with reduced adiposity (30) and the significant increases in both HP and HEP. Further analysis at the genus level showed that *Chlamydia*, *Bacteroides*,

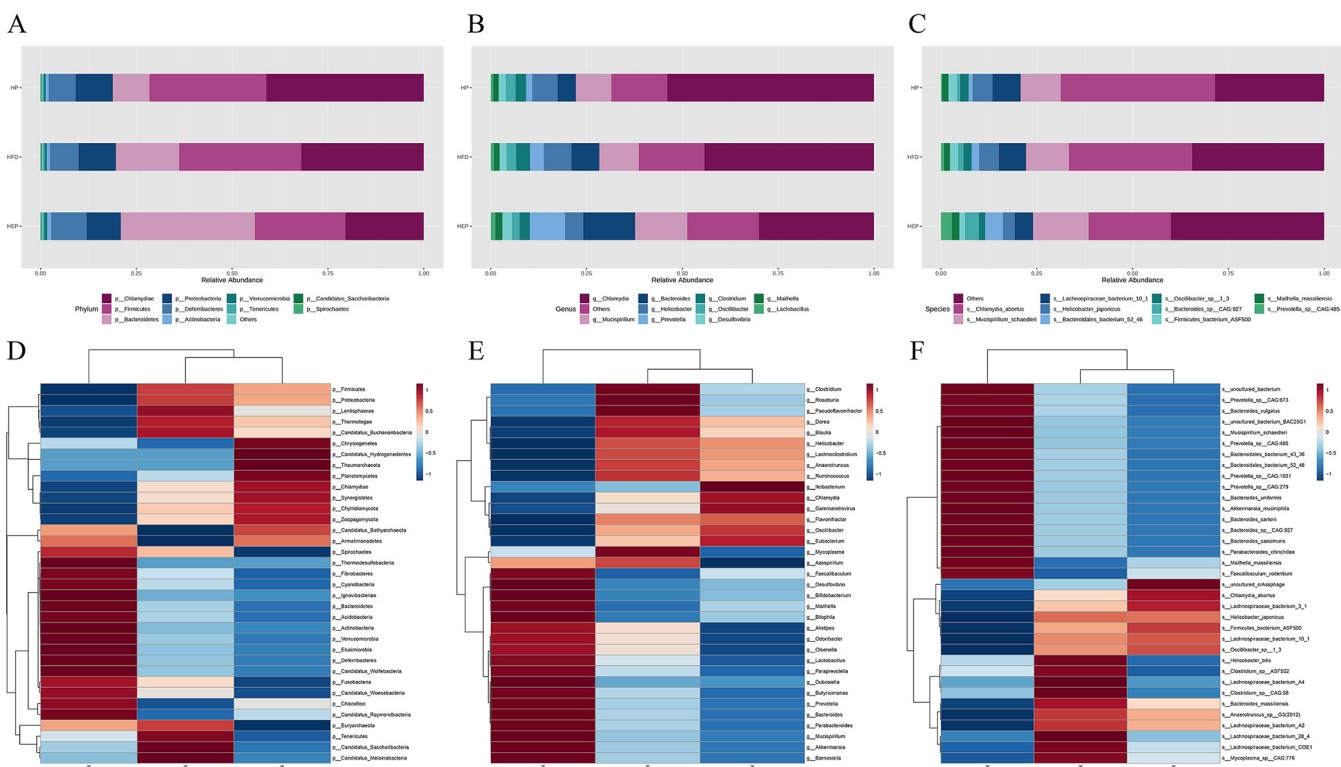

**FIG 3** Intestinal microbial classification composition of feces in each group. Relative abundance map of species taxonomic at the levels of phylum, genus and species: (A) phylum level; (B) genus level; (C) species level. Heatmap of fecal samples at the level of phylum, genus and species: (D) phylum level; (E) genus level; (F) species level.

*Mucispirillum*, *Prevotella*, *Helicobacter*, and *Clostridium* were dominant in each group. Compared with the HFD group, the HEP showed a decrease in *Chlamydia* and *Helicobacter* but an increase in *Bacteroides*, *Prevotella*, and *Mucispirillum* ($P < 0.001$). At the species level, HFD showed significant changes in *Chlamydia abortus*, *Mucispirillum schaedleri*, *Lachnospiraceae bacterium* 10-1, *Helicobacter japonicus*, Bacteroidales bacterium 52_46, *Bacteroides* sp. CAG:927, and *Prevotella* sp. CAG:485 differed from HFD. Gut microbes were one of the important participants in host metabolic activity, which played an important role in the regulation of host energy metabolism and body fat synthesis and storage (31). It was shown that the engineered bacteria had a significant effect on the development of intestinal dominant bacteria in mice on high-fat diets.

To further screen the biomarkers, we next performed linear discriminant analysis effect size (LEfSe) analysis to elucidate the changes in gut flora after inulin and engineered bacteria treatment. According to the LEfSe analysis results, we identified significant differential bacterial composition among the HFD, HP, and HEP groups (Fig. S3). In the HFD group, *Spiroplasma cantharicola*, *Pseudomonas* sp. AU12215, *Bacillus gottheilii*, and *Coprococcus* sp. 43_8 were identified to be important. The most noticeable change occurred in the HP group. An obvious alteration in the microbiota was characterized at the family level by *Chlamydiaceae* and Tolypothrichaceae in HP mice. At the genus level, the most significant changes were observed in *Tolypothrix*, *Roseivirga*, and *Gimesia*, among others. In addition, some species were more abundant in the HP group, such as *Yangia* sp. CCB-MM3, *Tolypothrix bouteillei*, and *Thermus amyloliquefaciens*, etc. Most importantly, we found higher Odoribacteraceae levels in the HEP group, with *Streptococcus caviae* and *Firmicutes bacterium* M10 2 as two key species. Among them, Odoribacteraceae could regulate glucose and lipid metabolism by degrading the concentration of succinic acid in the blood, whereas *Streptococcus* participated in host metabolism (28, 32). From the above results, it could be seen that the key species in the intestinal flora of mice differed after the intervention of inulin and engineered bacteria, indicating that there were differences in

the enrichment of key species in the intestinal flora of mice on a high-fat diet by inulin and engineered bacteria.

**Functional analysis of metagenomic sequencing revealed the effect of engineered bacteria on the function of intestinal flora.** Functional annotations can further help to sort out the difference phenotypes of metabolism between microorganisms and hosts and analyze their interaction process, which has an important relationship with the composition of the flora. We selected the Kyoto Encyclopedia of Genes and Genomes (KEGG) database and the carbohydrate-active enzyme (CAZy) database for functional annotation analysis. According to the KEGG pathway, there were six main biological metabolic pathways, including Cellular Process, Environmental Information Process, Genetic Information Process, Metabolism, Human Diseases, and Organismal Systems. As shown in Fig. 4A to C, the relative abundance of metabolic functional modules was significantly higher in the HEP group compared to the HFD group ($P < 0.001$), whereas the abundance of metabolic pathways such as carbohydrate metabolism, amino acid metabolism, metabolism of cofactors and vitamins ($P < 0.001$), and energy metabolism ($P < 0.01$) were significantly enhanced in HEP. For instance, the metabolic pathway activity of lipids, amino acids, vitamins, xenobiotic biodegradation, nucleotides, glycans, carbohydrates, and energy were higher in HEP than in HFD, whereas the opposite effect was observed in HP. Moreover, we compared the functional bacteria genes across the three groups according to the carbohydrate-active enzyme (CAZy) database and constructed a histogram for the main six major functional enzyme systems, including glycoside hydrolases (GHs), glycosyl transferases (GTs), polysaccharide lyases (PLs), carbohydrate esterases (CEs), auxiliary activities (AAs), and carbohydrate-binding modules (CBMs). As shown in Fig. 4D to F, there was variability in the functional modules of the gut flora among the experimental groups.

The relative abundance of glycoside hydrolases was significantly higher in HEP than in HFD ($P < 0.01$). After long-term prevention of BsS-RS06551, the abundance of some enzymes, such as GT2, GH43, GH2, GH97, GH3, and CE12, were significantly increased. These results suggested that these carbohydrate enzymes might be important in preventing obesity induced by a high-fat diet. The existence of variability in carbohydrase genes among the groups suggested that the alteration of intestinal flora by inulin and engineered bacteria affected the expression of carbohydrase genes, which in turn regulates the function of the flora.

**Variability in fecal metabolites and metabolic pathways revealed that the engineered bacteria affected the host metabolic profile.** Diet is an important factor affecting the metabolism, and high-fat diets can cause metabolic disorders and obesity (33, 34). Increasing evidence suggests that gut microbiota plays a key role in host metabolism related to obesity, and microbe-derived metabolites can influence the hosts via multiple pathways, such as physiology and behavior (17, 35). We next investigated the effect of inulin or BsS-RS06551 on the host metabolic profile following previously published methods (24). Principal-component analysis (PCA) and orthogonal partial least-squares discriminant analysis orthogonal partial least-squares discriminant analysis (OPLS-DA) showed different metabolomic profiles among the HP, HEP, and HFD groups (Fig. 5A and B). According to the results in the OPLS-DA model (variable importance in projection [VIP] score greater than 1; $P \leq 0.05$), a total of 168 characteristic metabolic molecules were filtered and visualized using the hierarchical clustering to cluster samples (Table S4). The results were displayed in a heat map and a tree diagram, respectively. It was found that the abundance of metabolites differed greatly between different mice, the difference within HFD group was especially large, and the metabolite levels in the HEP and HP groups were relatively concentrated (Fig. 5C and D). Metabolite identification started with confirmation of the exact molecular weight of the metabolite, followed by further matching of annotations in the standards database based on fragment information obtained from tandem mass spectrometry (MS/MS) patterns to obtain accurate information on the metabolite, and the identification profiles of some compounds are shown in the supplemental materials (supplemental file 1).

Next, these differential metabolites were mapped to the KEGG pathway via metabolite set enrichment analysis (MSEA) for investigating changes in metabolic pathways. The identified metabolites were annotated to 52 KEGG pathways (Fig. S4; Table S5), including the

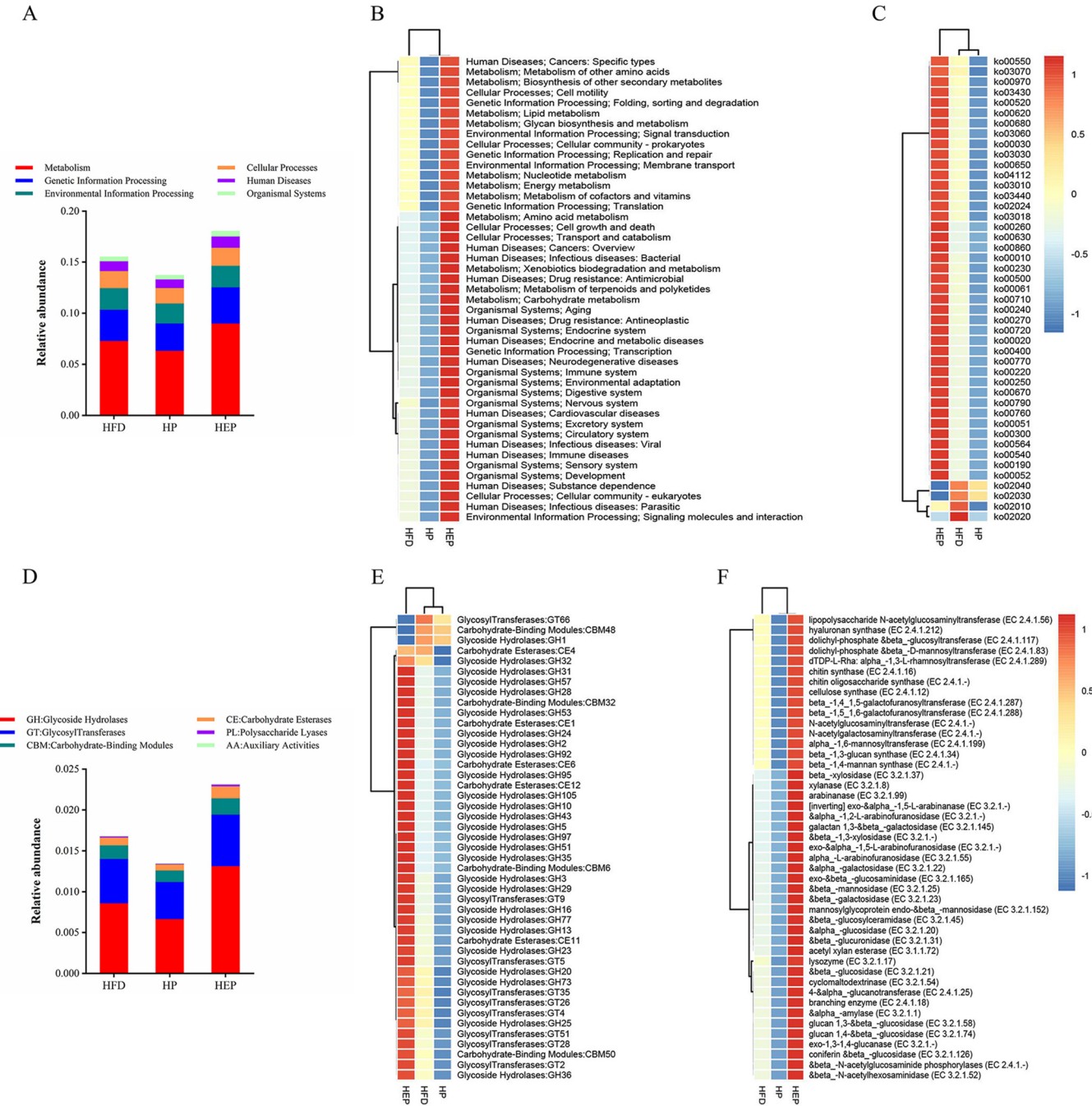

**FIG 4** Functional relative abundance of intestinal flora based on the KEGG database. (A) Histogram of the relative functional abundance of KEGG level 1. (B) Heat map of metabolic function clustering of KEGG level 2. (C) Heat map of metabolic function clustering of KEGG level 3. Functional relative abundance of intestinal flora based on the CAZy database. (D) Histogram of the relative functional abundance of CAZy level 1. (E) Heat map of metabolic function clustering of CAZy level 2. (F) Heat map of metabolic function clustering of CAZy level 3. GH, glycoside hydrolase; GT, glycosyl transferase; CBM, carbohydrate-binding modules; CE, carbohydrate esterases; PL, polysaccharide lyase; AA, auxiliary activities.

arginine biosynthesis pathway; histidine metabolism pathway; phenylalanine, tyrosine, and tryptophan biosynthesis pathway; valine, leucine, and isoleucine biosynthesis pathway; and primary bile acid biosynthesis pathway. To further analyze changes in metabolic pathways with regard to obesity and the intervention effects of engineered bacteria, we used functional meta-analysis to identify consistent functional changes by integrating three sets of data in pairs. Based on the results, the most pronounced differences were found in 14 KEGG pathways (Fig. S5; Table S6), including primary bile acid biosynthesis, steroid hormone biosynthesis, and biosynthesis of unsaturated fatty acids. Based on the above-mentioned

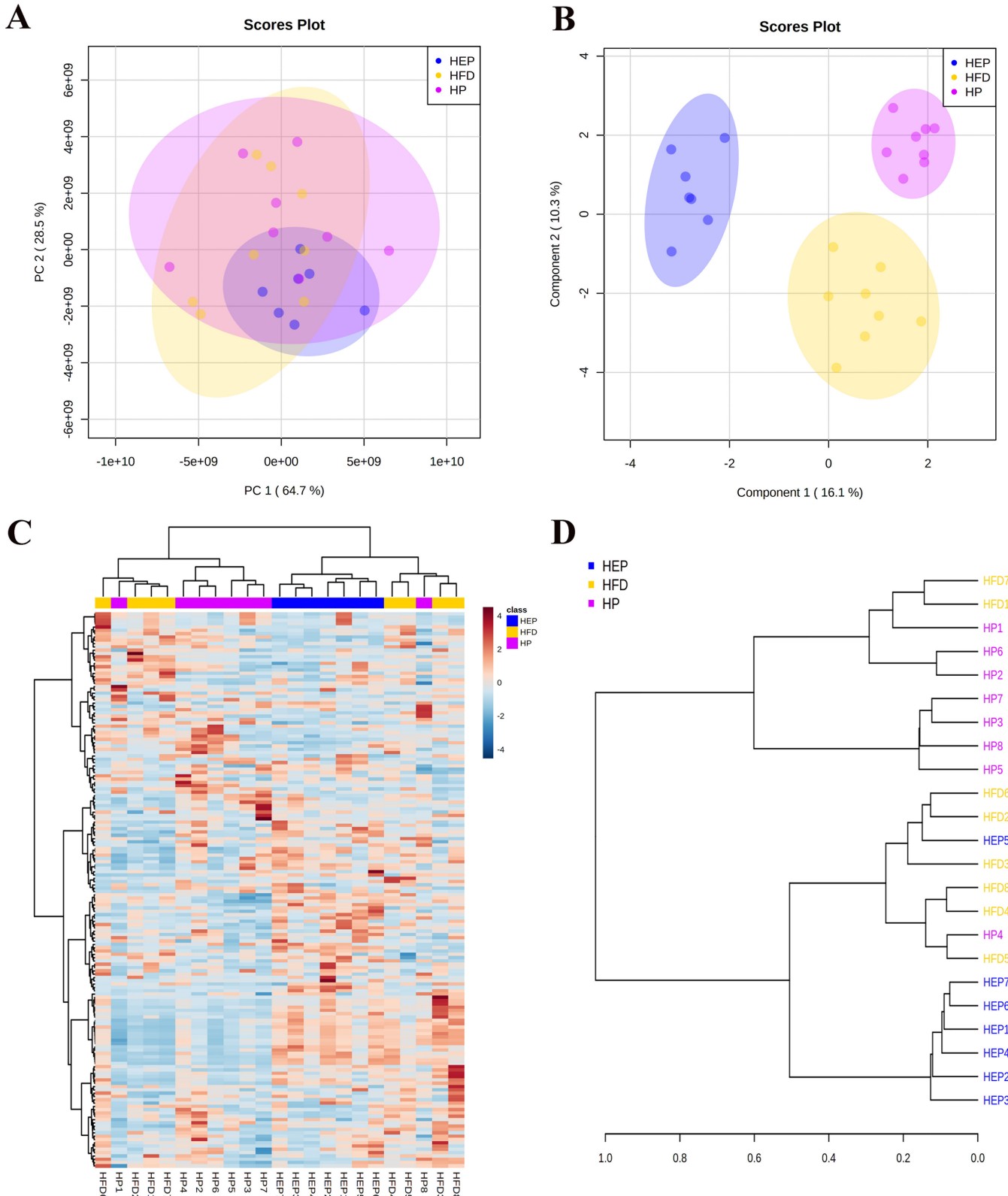

**FIG 5** Untargeted fecal metabolomics analysis. (A) Principal-component analysis (PCA) score plots of fecal metabolic profiles. (B) The sparse partial least-squares discriminant analysis (PLS-DA) score plots of fecal metabolic profiling of HFD, HP, and HEP. (C) Hierarchical clustering of differentially metabolites in HFD, HP, and HEP. (D) Tree analysis of all samples.

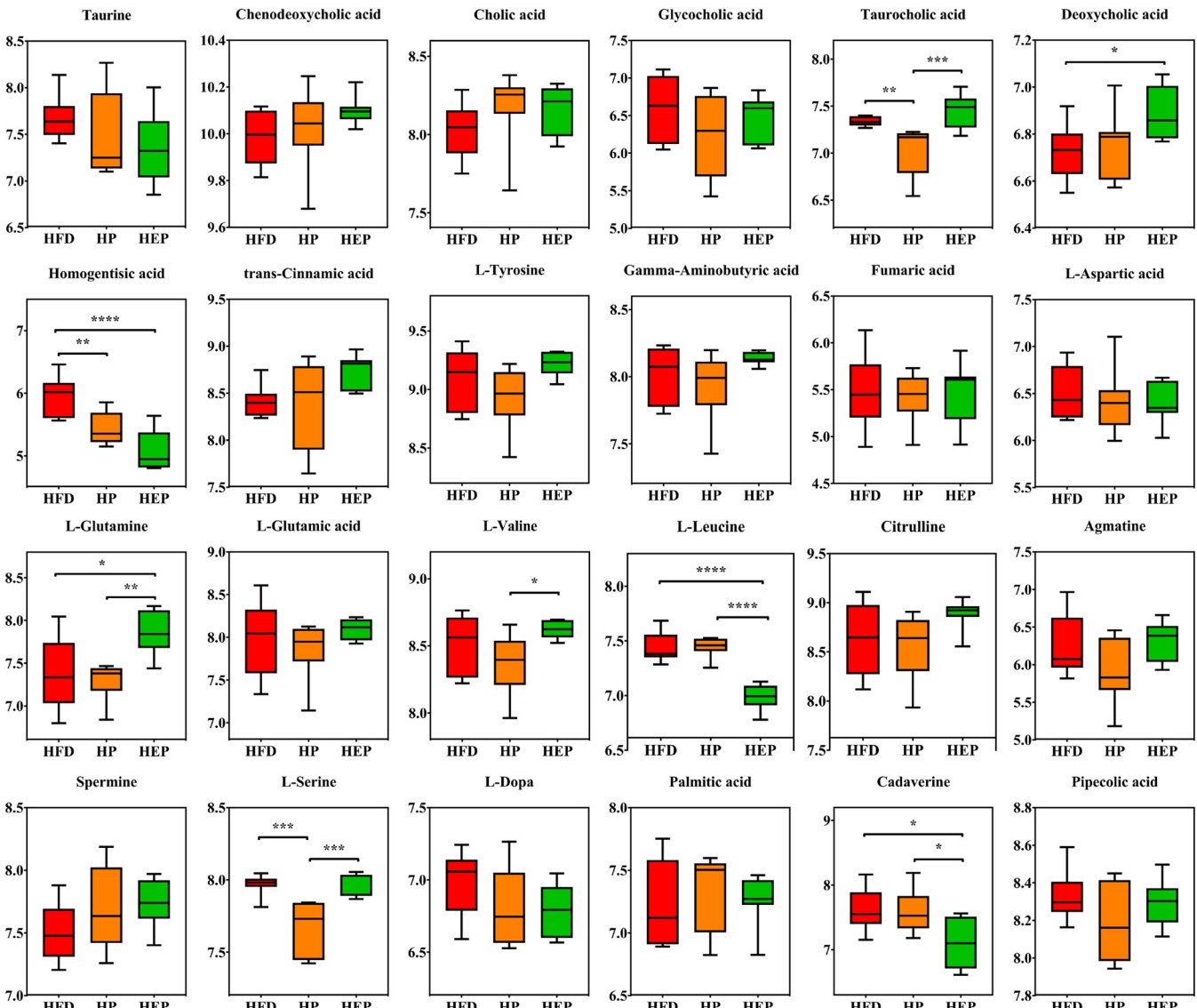

**FIG 6** Box plots of relative abundance of significance metabolites. HFD, HFD + inulin (HP), and HFD + BsS-RS06551 (HEP) groups (*n* = 8 for each group). The normalized intensity data with log function conversion (based on 10) to plot. *, *P* value < 0.05, **, *P* < 0.01, ***, *P* < 0.001. L-DOPA, L-3,4-dihydroxyphenylalanine.

analysis results, 24 significant differences in metabolites among the HFD, HP, and HEP groups were further analyzed (Fig. 6). After a 14-week intervention period with engineered bacteria in obese mice, we observed a significant decrease in taurine and glycocholic acid associated with bile acid metabolism pathways and an increase in the levels of chenodeoxycholic acid, cholic acid, and deoxycholic acid. HP showed a similar trend to HEP, except for taurocholic acid. Within some amino acid-related pathways, some important metabolites were annotated to more than one pathway. Compared with the obese mice, HEP groups showed a noticeable intervention effect with a significant reduction in homogentisic acid, L-leucine, L-3,4-dihydroxyphenylalanine (L-DOPA), and cadaverine. In contrast, *trans*-cinnamic acid, L-tyrosine, L-glutamine, L-valine, citrulline, agmatine, and spermine significantly increased. Further, γ-aminobutyric acid, fumaric acid, L-aspartic acid, L-glutamic acid, and palmitic acid also showed elevated or decreased levels.

To explore the possible relationship between altered gut microbiota composition and metabolites, we performed Spearman correlation analysis (Fig. 7). In the previous analysis results, 27 metabolites were selected as representatives in the difference changes of metabolic pathways, and 21 genera were selected as representatives in the difference changes of species

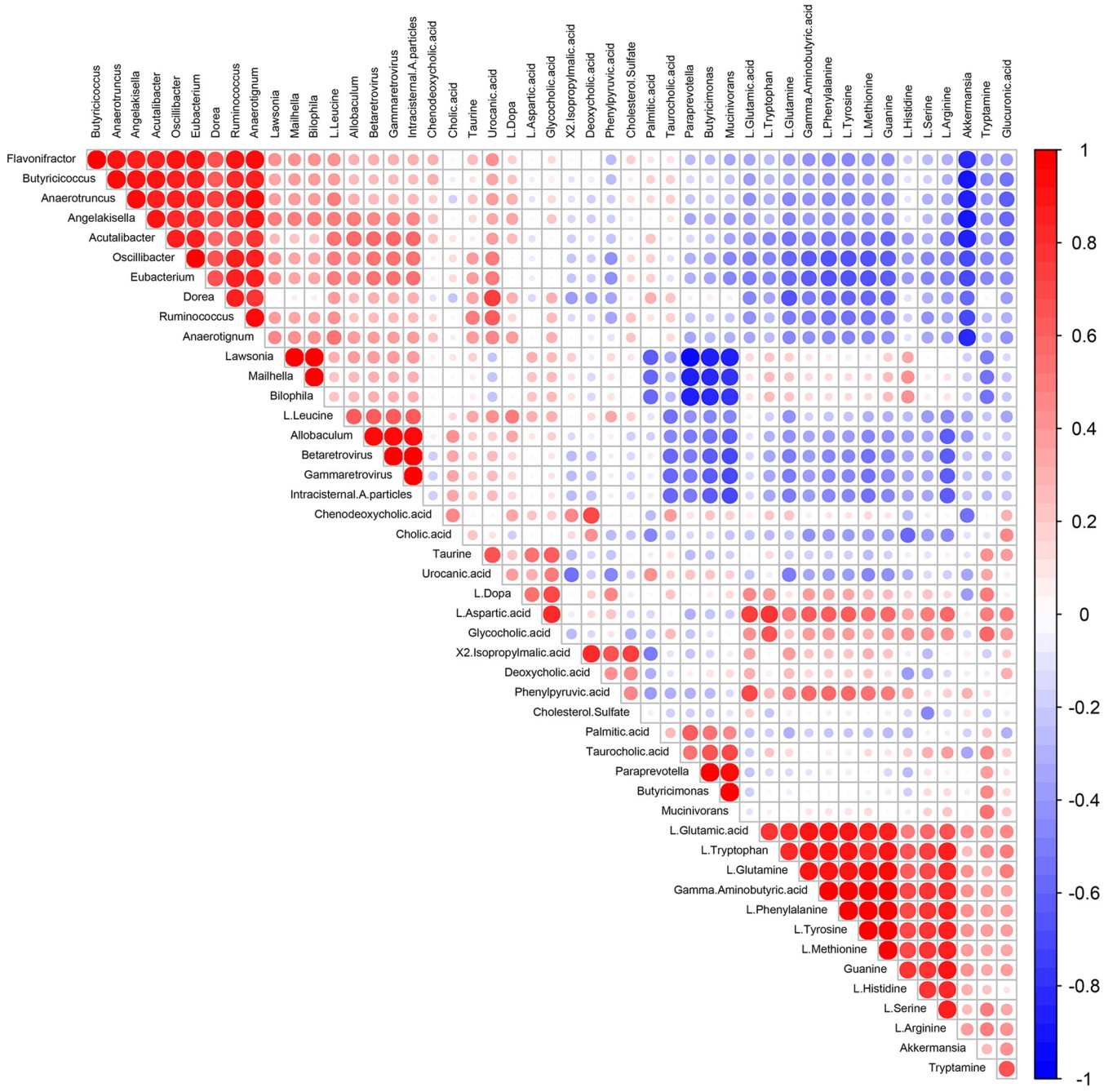

**FIG 7** Correlation plot of Spearman's correlation of fecal metabolites and genus.

for correlation analysis. We compared the 27 key fecal metabolites with selected 21 genera and visualized the heat map results according to the aforementioned experimental results and reported studies (Table S7) (18, 36). We found that some metabolites were the amino acids and their intermediate metabolites, especially L-Leucine, L-tryptophan, L-glutamine, and L-histidine. They were positively correlated with *Desulfovibrionaceae*, including the genera *Lawsonia*, *Mailhella*, and *Bilophila*. Notably, a similar trend was observed in *Akkermansia*. However, negative correlations were observed in Ruminococcaceae, *Flavonifractor*, *Butyricicoccus*, *Oscillibacter*, *Eubacterium*, and *Dorea*. Negative associations were also noted between L-leucine and genera *Paraprevotella*, *Butyricimonas*, *Mucinivorans*, and *Akkermansia*.

## DISCUSSION

As one of the beneficial active small molecules in the intestinal tract, butyrate is involved in the metabolism of sugars and lipids (37, 38), promoting the secretion of insulin and reducing appetite (39); this plays an important role in the gut host-microorganism metabolic axis (36). We constructed an engineered butyrate-producing strain, BsS-RS06551, using *B. subtilis* SCK6 as our cellular chassis and evaluated its therapeutic effects on obesity. The metabolites of the gut microbiota act as a medium for the interaction of the host and microorganisms, affecting human physiology and pathology. As a functional output of the interaction between the host and microorganisms, metabolites provide us with a timely snapshot of the complex multibiological system (40). Fecal metabolomics could intuitively analyze changes in the intestinal metabolism and metabolic pathways targeting the interaction between host and microorganisms (41, 42) and could be used to research many chronic intestinal diseases (43). Obesity is a chronic metabolic disease and has been thought to be associated with glucose and lipid metabolism disorders. The goal of our study was to evaluate the therapeutic interventions of butyrate-producing bacteria in obesity.

We summarized the possibilities to explain the positive effect of BsS-RS06551 on obese mice with regard to microbiota-gut-brain communication (Fig. 8). Alteration in the composition of gut microbiota with disturbed food intake may lead to obesity by causing a disorder in the brain and substantially modifying the host's metabolism (44, 45). Our intervention strategies with engineered butyrate-producing bacteria have a large probability of remodeling gut homeostasis, changing the composition and regulating the function of the intestinal flora. Previous studies have shown that butyrate has the ability to cross the blood-brain barrier and activate the vagus nerve and hypothalamus, thus indirectly affecting the appetite and feeding behavior of the host (46, 47). We found that the experimental results showed an increase in dietary intake but a decrease in body weight in mice under the intervention of butyric acid-producing engineered bacteria, suggesting that the engineered bacteria play a role in alleviating obesity while promoting energy absorption, and the mechanism may be related to the regulation of glucolipid metabolism by butyric acid through the intestine-brain axis. Moreover, recent studies have shown that levels of neurotransmitters, such as γ-aminobutyric acid, could regulate central appetite and food reward signaling by direct stimulation of the vagus nerve or indirect neuroendocrine and immune effects (48). We found that many differential metabolites were involved in neurotransmitter metabolic network, such as phenylalanine, tyrosine, tryptophan, histidine, aspartate, and glutamate metabolism, which suggested the involvement of intestinal flora in the synthesis and metabolism of neurotransmitters and implied the possibility that engineered bacteria influenced the increased metabolic benefits of neurotransmitters. Some beneficial metabolic effects of butyrate are mediated through gluconeogenesis in the intestinal epithelium and gut-brain neural circuits to increase insulin sensitivity and glucose tolerance (44, 49). Simultaneously, the metabolites generated by the interaction between host and gut microbiota could induce intestinal gluconeogenesis and thereby modulate host intestinal homeostasis. These metabolites mainly involved amino acid metabolism pathways, such as those associated with the metabolism of alanine, aspartate, glutamate, valine, leucine, phenylalanine, tyrosine, and tryptophan. Increased circulating levels of branched-chain amino acids (BCAAs) and aromatic amino acids (AAAs) were considered to be related to obesity (50, 51). Our engineered bacteria could cause a significant reduction in these amino acids and modulate their metabolite concentrations to some extent. Additionally, studies have been reported that bile acids are associated with obesity via the farnesoid X receptor (FXR) or Takeda G receptor 5 (TGR5) (52–55). The modulatory effect of the engineered bacteria on bile acids and their metabolites was observed in our results, including primary and secondary bile acid biosynthesis pathways. We found these obvious differences in gut microbiota and microbe-derived metabolites between different groups, indicating the beneficial effects of engineered bacteria in modulating the gut-brain axis, thereby enhancing insulin action to normal levels and targeting to prevent and counteract obesity. Studies have confirmed the strong association between intestinal flora and obesity, but the causal association and the underlying mechanisms are not yet clear (56). We found that the engineered bacteria

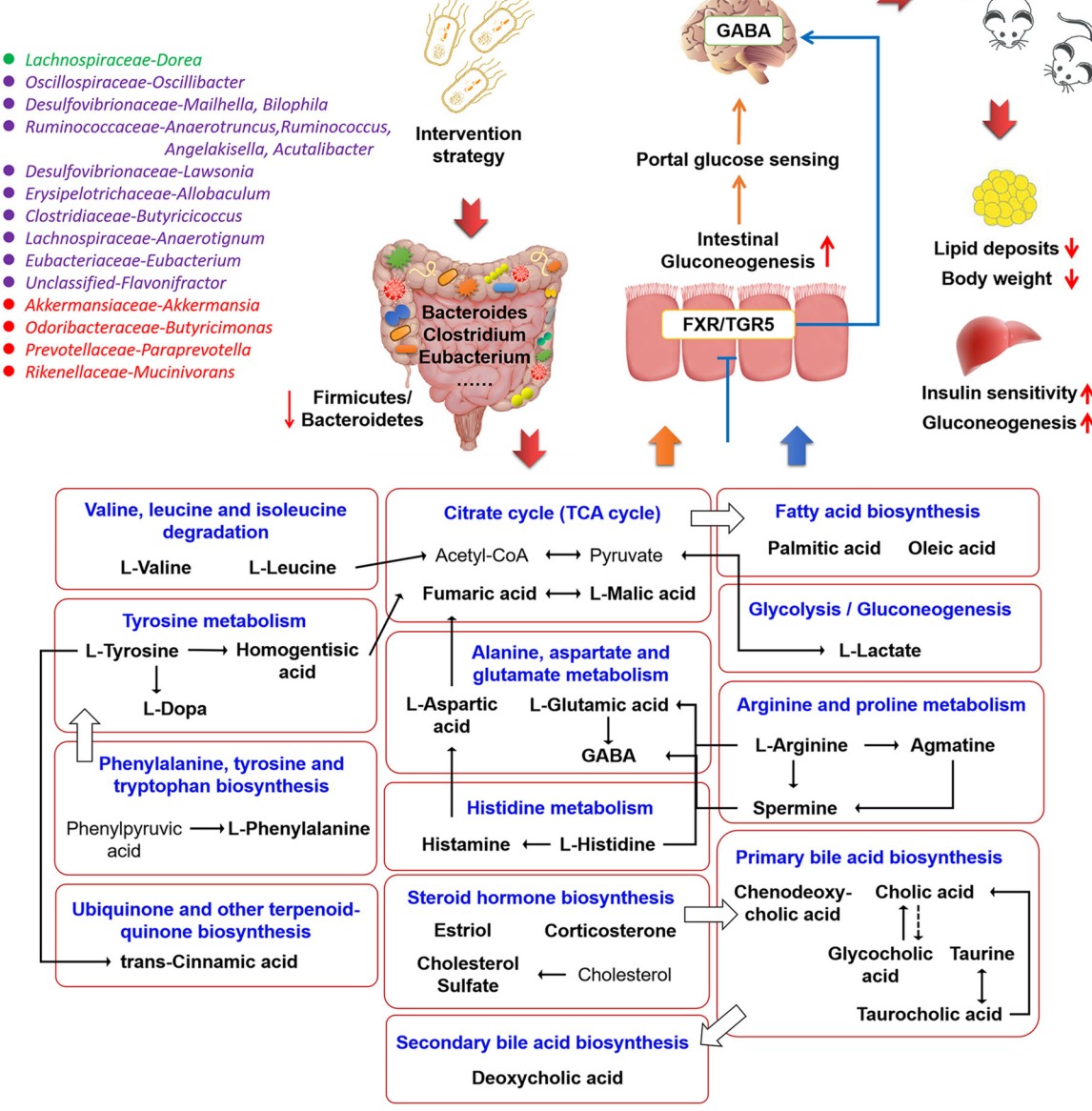

**FIG 8** The summary of the possibilities to explain the positive effect of BsS-RS06551 on obese mice toward microbiota-gut-brain communication.

could increase the abundance of beneficial bacteria, such as *Bifidobacterium*, *Lactobacillus*, and *Akkermansia* and decrease the relative abundance of the bacterial genera related to obesity, such as *Chlamydia* and *Helicobacter*. In a word, supplementation of engineered bacteria could change the structure of intestinal flora in mice and play a positive role in preventing obesity and improving metabolic function.

This study showed that long-term consumption of BsS-RS06551 had a significant inhibitory effect on obesity induced by a high-fat diet and was more potent in inhibiting obesity than prebiotic inulin. Furthermore, BsS-RS06551 showed a beneficial effect on host glucose metabolism, lipid metabolism, and gut microbe composition. However, the present work has some limitations; fecal metabolic analysis cannot fully reveal the true metabolic status of the host, and further studies are needed to confirm the association between gut microbial-mediated metabolites and obesity. This study provided insights into the relationship between the engineered bacteria on the host fecal microbiome, metabolites, and obesity and may be important for studying the mechanisms of the microbiota-gut-brain axis in obese patients. Further studies are needed to

confirm the relationship between gut microbial-mediated metabolites and the central nervous system. This paper presents possible future approaches for targeting engineered bacterial interventions associated with obesity metabolism. In addition, considering the colonization potential of bacteria (57, 58), the engineered bacteria reported here can provide a new strategy for the effective and convenient treatment of obesity in the long term.

## MATERIALS AND METHODS

**Strains, plasmids, and growth conditions.** The bacterial strains and plasmids used in this study are listed in Table S1. *Escherichia coli* DH5$\alpha$ was used as a host for cloning vectors. *B. subtilis* SCK6 was the original strain used in the genome engineering procedures. Plasmids and genomic DNA were extracted using the kit (TIANGEN) according to the manufacturer's instructions. PCR used the polymerase *Taq* (Vazyme). Restriction endonucleases and T4 DNA ligase were purchased from Thermo Scientific. During strain construction, the culture was grown aerobically in Luria-Bertani (LB) liquid medium (1% [wt/vol] trptone, 0.5% [wt/vol] yeast extract, 1% [wt/vol]) NaCl and LB agar plates at 30 or 37°C. Ampicillin (100 mg/liter), kanamycin (50 mg/liter), or spectinomycin (50 mg/liter) were used to select suitable strains. L-Arabinose was added for induction at a concentration of 1%, and the isopropyl-$\beta$-D-thiogalactoside (IPTG) was used at concentration of 0.5 mmol/liter.

**BsS-RS06551 construction.** The CRISPR/Cas9 genome editing system was used for *B. subtilis* SCK6 genome editing, which contained two important components: a CRISPR-associated endonuclease (Cas protein) and a guide RNA (gRNA) in plasmids pCas and pTargetF (59). The pCas was mainly used to produce the protein Cas9, which could cut the region of interest. The pTargetF contained important parts: a modified promoter P43, the target site N20 (20 bases DNA), the single guide RNA (sgRNA) sequence, and donor fragments for repairing double-strand breaks (a host-specific guide skeleton, homologous arm fragment according to the host specificity, and targeted gene information). An online software Cas-OFFinder (http://www.rgenome.net/) was used to design N20 with different editing requirements. Series of plasmids, pTargetF P1 to P4, were obtained according to the need of knocking out the target genes *skfA*, *sdpC*, *acdA*, and *ackA*. The plasmid pTargetF-P5 was obtained according to the requirement of knock-in target gene BCoAT. A series of primers was designed to verify gene deletions and insertions, as well as homologous recombination. The primer and N20 sequences used in this study are given in Table S2 in the supplemental material. For gene knockout, we transformed pCas and pTargetF into SCK6, spread the transformation solution onto LB plates containing two antibiotics (kanamycin and spectinomycin), cultivated overnight at 30°C, and further screened to obtain strains containing pCas and pTargetF. Then, L-arabinose was used to induce the bacteria for 16 h at 30°C and 200 rpm. The genome was extracted, and PCR and sequencing verification were performed. Following this, the sample was cultured with IPTG at 30°C, with continuous shaking at 200 rpm for 8 to 16 h to induce the elimination of pTargetF, and the resulting solution was spread on the LB plate containing kanamycin antibiotics for spectacular antibiotic resistance verification. Subsequently, a new pTargetF plasmid was introduced for the next round of gene editing. After editing, the pTargetF plasmid-eliminating bacteria were cultured on a shaker at 37°C and 200 rpm to eliminate pCas and diluted and spread on an LB plate without antibiotics for kanamycin resistance verification.

**Fermentation culture of BsS-RS06551 and determination of butyric acid content.** The constructed engineered bacteria were inoculated in 0.5-liter fermentation flasks with a starting inoculum of 1%; i.e., 5 mL of seed solution, and fermented in shaking bed at 37°C and 200 rpm for about 24 h. The pH value of the fermentation broth was monitored by pH electrode during the fermentation process. The content of butyric acid in the fermentation broth was determined by gas chromatography-mass spectrometry (GC-MS) at the end of fermentation. The detailed GC-MS process is shown in the supplemental materials.

**Animals and treatment regime.** Twenty-four male C57 BL/6J mice (3 weeks old) were randomly divided into three groups after 1-week acclimation. The mice had free access to water and food during the acclimation. The room was maintained at a temperature of 22 $\pm$ 3°C and relative humidity of 55% $\pm$ 10%, and the lighting regime was a standard 12-h:12-h light dark cycle, with lights from 8 a.m. to 8 p.m. After 1 week of acclimation, the mice were maintained on a high-fat diet for 14 weeks. The high-fat diet used a purified formula feed with 4.73 kcal/g of energy, of which 45% is provided by fat (20% by protein and 35% by carbohydrates) (Medicience Ltd.). The mice were randomized into three groups ($n = 8$ mice/group): mice gavage with 200 $\mu$L distilled water per day (HFD); mice gavage with 200 $\mu$L inulin solution (0.5 g/kg) per day (HP); and mice gavage with 200 $\mu$L bacteria solution ($10^8$ CFU/mL) per day (HEP).

**Physiological measurements.** Body weight and food intake were recorded once per week. Food was weighed before adding to individual mice cages three times a week. The leftover and spilled food was also weighed. Food intake was calculated after accounting for the spillage.

**Glucose tolerance, insulin tolerance, serum analysis, and liver histology.** The tests of glucose tolerance and insulin tolerance were performed as previously described (24). Fasting glucose levels were determined using the Mindary BS-2000M instrument (Shenzhen, China) according to the manufacturer's instructions. Fasting insulin resistance levels were determined using commercial insulin enzyme-linked immunosorbent assay (ELISA) kits. Insulin resistance was estimated using HOMA-IR (50). Plasma free acid levels were determined using a quantification kit (WAKO). The liver histology was analyzed as previously described (24).

**Untargeted LC-MS/MS for measurement of metabolites in fecal samples.** The untargeted LC-MS/MS analysis was performed as previously described (24). A total of 100 mg of feces was added to 500 $\mu$L of double-distilled water (ddH$_2$O) (4°C) and mixed thoroughly for 1 min; then 1 mL of methanol was

added (−20°C) for 1 min. The samples were processed in an ultrasonic machine for 10 min at room temperature and placed on ice for 30 min. This was followed by centrifugation at 14,000 rpm for 10 min at 4°C, and 1,200 $\mu$L of supernatant was concentrated. The samples were dissolved with 400 $\mu$L of 2-chlorophenylalanine (4 ppm) in methanol (1:1, 4°C) and filtered through a 0.22-$\mu$m membrane to obtain the samples for LC-MS. Chromatographic separation was accomplished in an Thermo Ultimate 3000 system, and the electrospray ionization mass spectrometry (ESI-MS$^n$) experiments were executed on the Thermo Scientific Q Exactive Focus hybrid quadrupole-Orbitrap mass spectrometer, enabling highly selective quantitative and qualitative analysis. Proteowizard software (version 3.0.8789) was used to convert the obtained raw data into mzXML format. The XCMS package of R (version 3.3.2) was used for peak identification, peaks filtration, and peak alignment, with the following main parameters: bw = 2, ppm = 15, peakwidth = c (5, 30), mzwid = 0.015, mzdiff = 0.01, and method = centWave. Afterwards, the data were batch-normalized for peak area. To detect biomarkers, the relative standard deviation of potential characteristic peaks in quality control (QC) samples (i.e., the coefficient of variation) must not exceed 30%. Principal-component analysis (PCA) and OPLS-DA were performed on the data matrix using the R package ropls. Hierarchical clustering maps of relative quantitative values of metabolites were obtained by the pheatmap program package in R (version 3.3.2). Hierarchical clustering is used to cluster all samples to form a tree diagram showing the similarity between samples using the clustering method: average - linkage. Consistent functional changes in metabolic pathways and metabolites were identified using the MetaboAnalyst 5.0 databases (60, 61) (https://www.metaboanalyst.ca/).

**Extraction and sequencing of metagenomic DNA.** Fecal samples were collected from all experimental mice beginning in the last week of 14 weeks. The collected fecal pellets were immediately transferred into liquid nitrogen and then stored at −80°C. Subsequently, the DNA was extracted using a FastDNA$^M$ SPIN kit for feces (catalogue no. 116570200, MP Biomedicals SARL, Illkirch, France) and processed according to the manufacturer's instructions. The extracted DNA was sonicated and randomly broken into fragments with lengths of approximately 300 to 350 bp, and then the base A and an adapter were added to the 3′ end of the DNA fragment to repair the modified DNA fragment. This was followed by purification and PCR amplification to complete the library preparation. Qubit 2.0 was used for preliminary quantitative analysis; the library was diluted to 2 ng/$\mu$L, and the insert fragment size of the library was detected using Agilent 2100. After reaching the preset values, quantitative PCR (qPCR) was performed to accurately quantify the effective concentration of the library to ensure optimum quality. Illumina PE150 sequencing was conducted after pooling different libraries according to the effective concentration and the target data volume (Novogene, China).

**Metagenomic sequence assembly and gene predictions.** Raw data obtained by Illumina HiSeq were preprocessed using Reasfq (V8, http://github.com/cjfields/readfq) to remove reads containing more than a certain percentage of low quality bases (less than or equal to 38%) (default value of 40 bp). We took out the reads with a certain percentage of N-terminal bases (default length was 10 bp) and removed the reads that overlap the adapter junction by more than 15 bp. The Bowtie2 software was used to filter raw data for host contamination, with the following parameters: end-to-end, sensitive, I 200, X 400. SOAP *de novo* (version 2.21) was used for assembly analysis to get scaftigs. USEARCH (version 7.0.1001) was used to reduce redundancy. SoapAligner (version 2.21) was used to obtain a filtered relative abundance table of scaftigs. Scaftigs no less than 300 bp were selected for gene prediction using MetaGeneMark (version 2.10), and CD-HIT (version 4.5.8) was used to cluster annotated protein sequences.

**Species and gene functional annotations.** Based on the MicroNR database, the obtained Uniqgenes were compared with the microbial sequence information in the database to obtain species annotation information for each gene (Unigene) and combined with the gene abundance tables to obtain species abundance tables for different taxonomic levels. Unigenes were compared and annotated with commonly used functional databases using DIAMOND (version 0.9.9) software: functional annotation and abundance analysis of metabolic pathways (KEGG) and carbohydrases (CAZy) from the gene catalogue (62).

**LEfSe analysis for significantly different species.** LEfSe was used to screen key species biomarkers to explain differences among all groups of mice (63). Differential abundance of species was tested on the database Galaxy (http://huttenhower.sph.harvard.edu/galaxy/).

**Statistical analysis.** The data are expressed as the means ± standard deviation and were subjected to one-way analysis of variance with one-way analysis of variance (ANOVA) Tukey multiple-comparison test and graphics presentation using GraphPad Prism software 8.0. Correlation analysis for assessing the relationship between gut microbiota and metabolites was conducted via Spearman correlations (Hmisc packages in R).

**Data availability.** The raw Illumina sequence data have been deposited in the NCBI database under BioProject accession no. PRJNA731974.

## SUPPLEMENTAL MATERIAL

Supplemental material is available online only.
**SUPPLEMENTAL FILE 1**, PDF file, 3.4 MB.
**SUPPLEMENTAL FILE 2**, XLSX file, 0.02 MB.
**SUPPLEMENTAL FILE 3**, XLSX file, 0.1 MB.
**SUPPLEMENTAL FILE 4**, XLSX file, 0.01 MB.
**SUPPLEMENTAL FILE 5**, XLSX file, 0.01 MB.
**SUPPLEMENTAL FILE 6**, XLSX file, 0.01 MB.

## ACKNOWLEDGMENTS

This work was supported by grant 2019YFA0905600 from the National Key Research and Development Project and grant 19YFSLQY00110 from the Science and Technology Program of Tianjin, China.

Animal experiments have followed all guidelines regarding laboratory animals and were approved by the Tianjin Management Committee of Laboratory Animals in the Institute of Radiation Medicine of the Chinese Academy of Medical Sciences.

Conceptualization and design, He Huang, Xiaocang Cao; construction of engineered bacteria experiments, Liang Bai, Xiaoming Cheng; conducting animal trials, Liang Bai, Xiaoming Cheng., Mengxue Gao, Guangbo Kang; data analysis, Liang Bai, Lina Wang; writing and revision, Lina Wang, Liang Bai, Guangbo Kang; all authors were responsible for reading and approving the manuscript.

We declare no conflict of interest.

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
