## [Reviewer comments · Microbiology Spectrum]

Microbiology Spectrum

Positive Interventional Effect of Engineered Butyrate-Producing Bacteria on Metabolic Disorders and Intestinal Flora Disruption in Obese Mice

Lina Wang, Xiaoming Cheng, Liang Bai, Mengxue Gao, Guangbo Kang, Xiaocang Cao, and He Huang

Corresponding Author(s): He Huang, Tianjin University

Review Timeline:

Submission Date:	August 16, 2021
Editorial Decision:	October 9, 2021
Revision Received:	October 14, 2021
Editorial Decision:	November 19, 2021
Revision Received:	January 20, 2022
Editorial Decision:	February 16, 2022
Revision Received:	February 17, 2022
Accepted:	February 17, 2022

Editor: Cheng-Yuan Kao

Reviewer(s): The reviewers have opted to remain anonymous.

Transaction Report:

DOI: <https://doi.org/10.1128/Spectrum.01147-21>

October 9, 2021

Prof. He Huang
Tianjin University
Tianjin
China

Re: Spectrum01147-21 (The Positive Intervention Effect of Engineered Butyrate-producing Bacteria on Metabolic Disorders and Intestinal Flora Disruption in Obese Mice)

Dear Prof. He Huang:

Thank you for submitting your manuscript to Microbiology Spectrum. When submitting the revised version of your paper, please provide (1) point-by-point responses to the issues raised by the reviewers as file type "Response to Reviewers," not in your cover letter, and (2) a PDF file that indicates the changes from the original submission (by highlighting or underlining the changes) as file type "Marked Up Manuscript - For Review Only". Please use this link to submit your revised manuscript - we strongly recommend that you submit your paper within the next 60 days or reach out to me. Detailed information on submitting your revised paper are below.

Link Not Available

Sincerely,

Cheng-Yuan Kao

Journals Department
Reviewer comments:

Reviewer #1 (Public repository details (Required)):

illumina sequencing data to be deposited in NCBI (Bioproject Accession PRJNA731974).

Reviewer #1 (Comments for the Author):

Please see attached comments.

Reviewer #2 (Comments for the Author):

The manuscript is well written and the responses to the reviewers are generally convincing. My only concern is that in the metabolomics analysis, the accuracy of the compound identifications is not convincing to me. In this study, a threshold of molecular weight error <20 ppm was used and a set of 168 metabolites were thus annotated using such filter criteria. What kind of mass spectrometer (mass analyzer) did the author use? Was it TOF or orbitrap? Typically for "putative" metabolite ID using

MS1 without authentic standard compounds, the molecular weight error of <5 ppm is required to avoid high FDR. In the manuscript, since the author described that the LC-MS/MS was implemented for metabolite IDs. I suggest that the author provide the MS/MS spectra for those annotated metabolites mentioned in the manuscript.

Staff Comments:

Preparing Revision Guidelines

Please return the manuscript within 60 days; if you cannot complete the modification within this time period, please contact me. If you do not wish to modify the manuscript and prefer to submit it to another journal, please notify me of your decision immediately so that the manuscript may be formally withdrawn from consideration by Microbiology Spectrum.

Reviewer Comments to Author

The manuscript by Wang *et al.* examines the long-term effects of engineered butyrate-producing bacteria on high-fat diet-induced obesity mouse model. In addition to phenotypic analyses, fecal metabolomics and metagenomics were used to monitor the effects of prebiotic (inulin) and probiotic treatments on metabolism and gut microflora. Given the emerging important roles of microbial SCFAs, especially butyrate, in host physiology, the application and effects of butyrate-producing microbes as next-generation probiotics in the treatment and prevention of gut-associated diseases is of interest to the community. However, the manuscript in its current form has major issues in logic flow, presentation, and scientific rigor that needs to be carefully addressed. Specific comments and suggestions are detailed below:

Major Comments:

1. Part of the novelty of this study is the new engineered *B. subtilis* RS06551 strain but I am unsure how novel and how relevant this new strain is to this study.
 - a. How is this RS06551 different from RS06550 (PMID: 32334588)? Based on the available information, they have the deletions and insertion of the same genes. It is unclear what is different between the two.
 - b. How much improvement in butyrate production over the previously published RS06550? Is this improvement in butyrate production replicated *in vivo* or under *in vivo* simulated conditions?
 - c. Are there any evidence to that RS06551 performs better than RS06550 *in vivo*? With the lack of comparison, it is unclear from this paper if RS06551 is even needed for the study, or if the previously reported and better characterized RS06550 strain will perform similarly in this long-term obesity study.
2. Why is inulin included as the only control in all the experiments if the purpose of the study is to determine the “interventional effect” of engineered bacteria? In other words, can engineered bacteria treatment alleviate or reverse the effects of high-fat diet? What is normal if there is no control for normal diet?
3. Furthermore, if inulin is taken to be a “normal control”, how can RS06551 be “more potent in inhibiting obesity than prebiotic inulin” as claimed multiple times in this study? What are the evidence to support this claim?
4. There are additional claims in the paper without sufficient support. Please elaborate to support your following claims or modify them accordingly if they cannot be supported:
 - a. Line 95: According to the results, the engineered bacteria can not only restore the normal flora level and metabolic state of obese mice, ...
 - b. Line 202: Taken together, the above analyses implicated that obese mice showed dysbiosis in gut microecology, and inulin or engineered bacteria supplementation could prevent the imbalance of intestinal flora and modulate the intestinal microecology.
 - c. Line 206: Effects of engineered bacteria on restoring intestinal microbiota function revealed by functional analysis of metagenomic sequencing.
 - d. Line 235: This indicated that the intervention with engineered bacteria altered the amounts of carbohydrate enzyme genes and affected the microbial community, thereby regulating the function of the flora, which was beneficial for the prevention of obesity symptoms.
5. From a technical and reproducibility point, it is very difficult to understand, let alone replicate, how the experiments are performed in this study due to missing essential experimental details in the materials and methods. They should at the very least be included in supplemental materials.

- a. Construct details such as where the gene sequences were obtained, are they synthesized, codon optimized etc. the primers sequences, spacer sequences etc.
 - b. How was experiment to measure butyrate production conducted? Nothing was mentioned in manuscript on this.
 - c. H&E and oil red staining
 - d. Metabolomics: protocol (how metabolites were extracted), how the data was analyzed, filter criteria ... are missing
 - e. Metagenomic: metagenomic and metataxonomic data processing was vaguely written with just the software used listed without details on filtering, sequence removal (if any), how genus was assigned, statistical analysis etc.
 - f. Data for producing figures 3 and 4, as well as calculated p-values, should be included as supplementary tables.
 - g. In Figure 5D, describe how the tree was obtained.
 - h. Please provide vendors and catalog numbers for commercial kits used in this study for improved reproducibility.
6. Line 181: "difference...is more significant" is fundamentally incorrect.
 7. Line 220: what does levels 1 and 2 mean?
 8. Please clarify: In the methods section for extraction and sequencing of metagenomic DNA, "fecal pellets from all mice were collected at the end of the acclimatization week." In the methods section for animals and treatment regime, "...mice were randomly divided into three groups after one week of acclimation". How can the metagenomic data reflect differences of the treatments if the data was acquired prior to the treatment?
 9. The supplementary material is supposed to support the main manuscript but in its current form cannot be understood by readers.
 - a. All supplementary figure legends lack sufficient description to inform readers what they are looking at.
 - b. Supplementary figure 2 is too small to be legible.
 - c. Supplementary figure 3 by itself is meaningless. A better way is to incorporate the information into supplementary table 2 instead.
 - d. Supplementary figure 4 clearly shows pathways but the legend says "analysis of metabolites". More details have to be provided how the data presented is achieved.
 10. Although this is a follow-up study with highly similar approaches to what was published, the previous study (PMID: 32334588) was only referenced in this manuscript to escape the need to fully elaborate scientific methods. There is hardly any comparison of results of the two studies and a discussion of the comparison. What did we learn from long term probiotic intervention using engineered bacteria based that is new compared to the short-term study?
 11. It will be useful and interesting to reanalyze the metagenomics data to determine if the different treatments affect the diversity of the gut microbial communities.
 12. It will be great if this study may be placed in the context of what is done by the community. How do the metabolic and microbial compositional changes in the gut in this study differ or similar to studies involving butyrate supplementation or other high butyrate producing microbes?
 13. The discussion section needs to undergo major correction since large sections of it should be in the materials and methods or results sections. Better organization of information within the manuscript will help improve its logic flow.

Minor Comments:

1. Please correct the following:

- a. Line 77, “transparent genetic background”
 - b. “engineering bacteria” was used incorrectly as a noun multiple times in the manuscript.
 - c. Line 354: “Deferribactere”
 - d. Line 519. “Turkey’ test”
 - e. Line 422 “Ampicillin (100 mg/L) and kanamycin (50 mg/L)” – please check your concentrations.
2. Please provide references for the “colonization potential of bacteria”.

November 19, 2021

Prof. He Huang
Tianjin University
Tianjin
China

Re: Spectrum01147-21R1 (The Positive Intervention Effect of Engineered Butyrate-producing Bacteria on Metabolic Disorders and Intestinal Flora Disruption in Obese Mice)

Dear Prof. He Huang:

Please specifically address this: The raw Illumina sequence data to be deposited in the NCBI database under BioProject accession no. PRJNA731974.

Link Not Available

Sincerely,

Cheng-Yuan Kao

Journals Department
Reviewer comments:

Reviewer #1 (Public repository details (Required)):

The raw Illumina sequence data to be deposited in the NCBI database under BioProject accession no. PRJNA731974.

Reviewer #1 (Comments for the Author):

It appears that my comments from the first revision were not available to the author and thus were not addressed at all in this revision. I am reiterating the comments from the first revision below.

Reviewer Comments to Author

The manuscript by Wang et al. examines the long-term effects of engineered butyrate-producing bacteria on high-fat diet-induced obesity mouse model. In addition to phenotypic analyses, fecal metabolomics and metagenomics were used to monitor the effects of prebiotic (inulin) and probiotic treatments on metabolism and gut microflora. Given the emerging important roles of microbial SCFAs, especially butyrate, in host physiology, the application and effects of butyrate-producing microbes as next-

generation probiotics in the treatment and prevention of gut-associated diseases is of interest to the community. However, the manuscript in its current form has major issues in logic flow, presentation, and scientific rigor that needs to be carefully addressed. Specific comments and suggestions are detailed below:

Major Comments:

1. Part of the novelty of this study is the new engineered *B. subtilis* RS06551 strain but I am unsure how novel and how relevant this new strain is to this study.
 - a. How is this RS06551 different from RS06550 (PMID: 32334588)? Based on the available information, they have the deletions and insertion of the same genes. It is unclear what is different between the two.
 - b. How much improvement in butyrate production over the previously published RS06550? Is this improvement in butyrate production replicated in vivo or under in vivo simulated conditions?
 - c. Are there any evidence to that RS06551 performs better than RS06550 in vivo? With the lack of comparison, it is unclear from this paper if RS06551 is even needed for the study, or if the previously reported and better characterized RS06550 strain will perform similarly in this long-term obesity study.
2. Why is inulin included as the only control in all the experiments if the purpose of the study is to determine the "interventional effect" of engineered bacteria? In other words, can engineered bacteria treatment alleviate or reverse the effects of high-fat diet? What is normal if there is no control for normal diet?
3. Furthermore, if inulin is taken to be a "normal control", how can RS06551 be "more potent in inhibiting obesity than prebiotic inulin" as claimed multiple times in this study? What are the evidence to support this claim?
4. There are additional claims in the paper without sufficient support. Please elaborate to support your following claims or modify them accordingly if they cannot be supported:
 - a. Line 95: According to the results, the engineered bacteria can not only restore the normal flora level and metabolic state of obese mice, ...
 - b. Line 202: Taken together, the above analyses implicated that obese mice showed dysbiosis in gut microecology, and inulin or engineered bacteria supplementation could prevent the imbalance of intestinal flora and modulate the intestinal microecology.
 - c. Line 206: Effects of engineered bacteria on restoring intestinal microbiota function revealed by functional analysis of metagenomic sequencing.
 - d. Line 235: This indicated that the intervention with engineered bacteria altered the amounts of carbohydrate enzyme genes and affected the microbial community, thereby regulating the function of the flora, which was beneficial for the prevention of obesity symptoms.
5. From a technical and reproducibility point, it is very difficult to understand, let alone replicate, how the experiments are performed in this study due to missing essential experimental details in the materials and methods. They should at the very least be included in supplemental materials.
 - a. Construct details such as where the gene sequences were obtained, are they synthesized, codon optimized etc. the primers sequences, spacer sequences etc.
 - b. How was experiment to measure butyrate production conducted? Nothing was mentioned in manuscript on this.
 - c. H&E and oil red staining
 - d. Metabolomics: protocol (how metabolites were extracted), how the data was analyzed, filter criteria ... are missing
 - e. Metagenomic: metagenomic and metataxonomic data processing was vaguely written with just the software used listed without details on filtering, sequence removal (if any), how genus was assigned, statistical analysis etc.
 - f. Data for producing figures 3 and 4, as well as calculated p-values, should be included as supplementary tables.
 - g. In Figure 5D, describe how the tree was obtained.
 - h. Please provide vendors and catalog numbers for commercial kits used in this study for improved reproducibility.
6. Line 181: "difference...is more significant" is fundamentally incorrect.
7. Line 220: what does levels 1 and 2 mean?
8. Please clarify: In the methods section for extraction and sequencing of metagenomic DNA, "fecal pellets from all mice were collected at the end of the acclimatization week." In the methods section for animals and treatment regime, "...mice were randomly divided into three groups after one week of acclimation". How can the metagenomic data reflect differences of the treatments if the data was acquired prior to the treatment?
9. The supplementary material is supposed to support the main manuscript but in its current form cannot be understood by readers.
 - a. All supplementary figure legends lack sufficient description to inform readers what they are looking at.
 - b. Supplementary figure 2 is too small to be legible.
 - c. Supplementary figure 3 by itself is meaningless. A better way is to incorporate the information into supplementary table 2 instead.
 - d. Supplementary figure 4 clearly shows pathways but the legend says "analysis of metabolites". More details have to be provided how the data presented is achieved.
10. Although this is a follow-up study with highly similar approaches to what was published, the previous study (PMID: 32334588) was only referenced in this manuscript to escape the need to fully elaborate scientific methods. There is hardly any comparison of results of the two studies and a discussion of the comparison. What did we learn from long term probiotic intervention using engineered bacteria based that is new compared to the short-term study?
11. It will be useful and interesting to reanalyze the metagenomics data to determine if the different treatments affect the diversity of the gut microbial communities.
12. It will be great if this study may be placed in the context of what is done by the community. How do the metabolic and microbial compositional changes in the gut in this study different or similar to studies involving butyrate supplementation or other

high butyrate producing microbes?

13. The discussion section needs to undergo major correction since large sections of it should be in the materials and methods or results sections. Better organization of information within the manuscript will help improve its logic flow.

Minor Comments:

1. Please correct the following:

- a. Line 77, "transparent genetic background"
 - b. "engineering bacteria" was used incorrectly as a noun multiple times in the manuscript.
 - c. Line 354: "Deferribactere"
 - d. Line 519. "Turkey' test"
 - e. Line 422 "Ampicillin (100 mg/L) and kanamycin (50 mg/L)" - please check your concentrations.
2. Please provide references for the "colonization potential of bacteria".

Reviewer #2 (Comments for the Author):

The author failed to provide the MS/MS spectra of the annotated compounds but since the mass spectrometry details are provided the other experts may judgement the efficacy of compound IDs on their own.

Staff Comments:

Preparing Revision Guidelines

Please return the manuscript within 60 days; if you cannot complete the modification within this time period, please contact me. If you do not wish to modify the manuscript and prefer to submit it to another journal, please notify me of your decision immediately so that the manuscript may be formally withdrawn from consideration by Microbiology Spectrum.

We appreciate the time and effort the Editor and the Reviewers dedicated to providing feedback on our manuscript and are grateful for the insightful comments and valuable improvements to our work. We have tried our best to revise the manuscript according to the construction comments and suggestions by the Reviewers. Those changes are highlighted in the manuscript. We sincerely hope that the Editor can reconsider our revised manuscript for the journal. Our response to the reviewers' comments is as follows:

Reviewer: 1

RESPONSE: When we revised the manuscript for the first time, due to some unknown reasons and mistakes, your comments were not shown in the email we received or in the submission system, for which we are very sorry. We appreciate your scientific, objective and impartial comments and suggestions on our manuscript. Please find the following detailed responses to your comments and suggestions. We sincerely hope that this revised manuscript has addressed all your comments

Major comments:

1. Part of the novelty of this study is the new engineered *B. subtilis* RS06551 strain but I am unsure how novel and how relevant this new strain is to this study.
 - a. How is this RS06551 different from RS06550 (PMID: 32334588)? Based on the available information, they have the deletions and insertion of the same genes. It is unclear what is different between the two.
 - b. How much improvement in butyrate production over the previously published RS06550? Is this improvement in butyrate production replicated in vivo or under in vivo simulated conditions?
 - c. Are there any evidence to that RS06551 performs better than RS06550 in vivo? With the lack of comparison, it is unclear from this paper if RS06551 is even needed for the study, or if the previously reported and better characterized RS06550 strain will perform similarly in this long-term obesity study.

RESPONSE: Thanks for your professional comments and questions. Taking into account the need for the yield of the product, we constructed the new genetic engineering strain RS06551 with 1.5-fold higher butyric acid production compared to RS06550, which was not explicitly shown in

the manuscript. In addition, the difference between this study and the previous ones is the setting of mice experiments. In the previous study, we constructed a high-fat diet-induced obesity mouse model over 10 weeks, followed by a 4-week RS06550 gavage intervention. In this study, we hope to explore the long-term intervention effect of the genetic engineering strain on the internal environment of normal mice that were fed a high-fat diet. Thus, for 14 weeks, normal mice were fed a high-fat diet with RS06551 as the prevention group, and mice fed a high-fat diet alone as the model group. After that, we used the metagenomics and metabolomics strategies to discover the preventive effect of BsS-RS06551 on obese mice at the level of intestinal flora and metabolites, respectively. Moreover, the safety and effectiveness of the butyrate-producing bacteria were better than the original strain in short-term treatment, which has been demonstrated in the previous article. According to the results, the butyrate-producing bacteria can change the composition and structure of the intestinal flora and affect host metabolism, thus preventing obesity in long-term intervention. The effect of BsS-RS06551 was comparable to inulin and even superior to inulin phenotypically. According to *in vivo* data, we put forward the hypothesis that BsS-RS06551 can prevent obesity in mice. Due to the inner complexity of the organisms, further study is required to investigate the mechanism and interaction between altered gut microbiota and related metabolic disturbances targeting obesity. Research in these directions is underway. According to the question you raised, we have made the modification in line 106-125, 436-464.

- a. The main difference between the two strains is that we further knocked out the acetate competition pathway (including two genes *acdA* and *ackA*) and changed the position of the BCoAT insertion, which has been added in Table S1 and S2.
- b. Our previous engineered strain BsS-RS06550 yielded 1 g/L *in vitro*, while the yield of BsS-RS06551 with further genetic modifications in this study was increased by 1.5-fold.
- c. The positive effects of butyrate in obese mice and other models have been reported (doi: 10.3389/fimmu.2021.678360; 10.1016/j.immuni.2018.12.018; 10.1016/j.ebiom.2018.03.030, etc), but there are relatively few studies on butyrate-engineered bacteria in mice. In these articles, butyrate was administered at doses ranging from 50-300 mg/kg. We wanted to use engineered bacteria with a higher butyric acid production capacity to explore their long-term preventive effects on obesity. The question you mentioned about the performance of engineered

bacteria in vivo is also of interest to us. We are conducting research on the in vivo colonization and butyric acid production capacity of butyric acid engineered bacteria and hope to address this issue in a future article.

2. Why is inulin included as the only control in all the experiments if the purpose of the study is to determine the "interventional effect" of engineered bacteria? In other words, can engineered bacteria treatment alleviate or reverse the effects of high-fat diet? What is normal if there is no control for normal diet?

RESPONSE: Thanks for your professional comments and questions. We have previously researched many studies on the effects of inulin on improving health indicators, including many animal studies and human clinical trials (doi: 10.1038/s41598-017-06447-x; 10.1016/j.chom.2017.11.003; 10.1186/s12986-015-0033-2; 10.4093/dmj.2013.37.2.140; 10.1053/j.gastro.2017.05.055). Considering the effects that inulin has shown in reducing weight gain due to diet and improving insulin resistance due to diet, and as a control or intervention in many experiments, we chose inulin as the control group. Therefore, we chose to feed normal mice a high-fat diet with inulin as a control, hoping to investigate whether the engineered bacteria can achieve the same intervention effect on obesity as inulin, and to try to explain the differences between them and obese mice from the perspective of metabolites and flora.

3. Furthermore, if inulin is taken to be a "normal control", how can RS06551 be "more potent in inhibiting obesity than prebiotic inulin" as claimed multiple times in this study? What are the evidence to support this claim?

RESPONSE: Thanks for your professional comments and questions. As mentioned above, instead of grouping mice with inulin intervention as normal controls, we wanted to compare whether the effect of the engineered bacteria intervention was similar to or even better than that of inulin. Our obese mice model was established according to the previous method by feeding normal diet (SPF grade normal mouse chow with 3.85 kcal/g of energy and 10% of energy from fat (purchased from Beijing Viton Lever Laboratory Animal Technology Co.). The high-fat diet was a purified formula feed with 4.73 kcal/g of energy, of which 45% is provided by fat (20% by protein and 35% by carbohydrates, purchased from Jiangsu Medison Pharmaceutical Co.).

4. There are additional claims in the paper without sufficient support. Please elaborate to support your following claims or modify them accordingly if they cannot be supported:
 - a. Line 95: According to the results, the engineered bacteria can not only restore the normal flora level and metabolic state of obese mice, ...
 - b. Line 202: Taken together, the above analyses implicated that obese mice showed dysbiosis in gut microecology, and inulin or engineered bacteria supplementation could prevent the imbalance of intestinal flora and modulate the intestinal microecology.
 - c. Line 206: Effects of engineered bacteria on restoring intestinal microbiota function revealed by functional analysis of metagenomic sequencing.
 - d. Line 235: This indicated that the intervention with engineered bacteria altered the amounts of carbohydrate enzyme genes and affected the microbial community, thereby regulating the function of the flora, which was beneficial for the prevention of obesity symptoms.

RESPONSE: Thank you very much for your very professional comments and clear suggestions, which are very helpful for us to get scientific rigor of the manuscript.

- a. Based on your suggestions, we have made further corrections to tighten up the content we want to convey in line 98-100 “According to the results, the engineered bacteria could reduce weight gain caused by high-fat diets, improve glucose tolerance and insulin tolerance, alter the composition and function of the intestinal flora, and affect host metabolism”.
- b. According to your suggestions, we have made the modification in line 247-252. “From the above results, it could be seen that the key species in the intestinal flora of mice differed after the intervention of inulin and engineered bacteria, indicating that there were differences in the enrichment of key species in the intestinal flora of mice on a high-fat diet by inulin and engineered bacteria”.
- c. Based on your suggestions, we have made further corrections to tighten up the content we want to convey in line 251-252 “Functional analysis of metagenomic sequencing revealed the effect of engineered bacteria on the function of intestinal flora”.
- d. Based on your suggestions, we have made further corrections to tighten up the content we want

to convey in line 279-282 “The existence of variability in carbohydrase genes among the groups suggested that the alteration of intestinal flora by inulin and engineered bacteria affected the expression of carbohydrase genes, which in turn regulates the function of the flora”.

5. From a technical and reproducibility point, it is very difficult to understand, let alone replicate, how the experiments are performed in this study due to missing essential experimental details in the materials and methods. They should at the very least be included in supplemental materials.
 - a. Construct details such as where the gene sequences were obtained, are they synthesized, codon optimized etc. the primers sequences, spacer sequences etc.
 - b. How was experiment to measure butyrate production conducted? Nothing was mentioned in manuscript on this.
 - c. H&E and oil red staining
 - d. Metabolomics: protocol (how metabolites were extracted), how the data was analyzed, filter criteria ... are missing
 - e. Metagenomic: metagenomic and metataxonomic data processing was vaguely written with just the software used listed without details on filtering, sequence removal (if any), how genus was assigned, statistical analysis etc.
 - f. Data for producing figures 3 and 4, as well as calculated p-values, should be included as supplementary tables.
 - g. In Figure 5D, describe how the tree was obtained.
 - h. Please provide vendors and catalog numbers for commercial kits used in this study for improved reproducibility

RESPONSE: Thank you very much for your very constructive comments and professional suggestions, which is valuable for improving the accuracy of the manuscript.

- a. Based on your suggestions, we have revised the method and some detailed information about the construction of the engineered strains in the MATERIALS AND METHODS section in line 436-464, which was also added to the attached materials.

- b. Based on your suggestions, we revised the Materials and MATERIALS AND METHODS section in line 465-472 and specified the method of determination of butyric acid content in the supplementary material.
 - c. Based on your suggestions, we have included the H&E and Oil Red staining methods in the supplementary material.
 - d. We thank the reviewer for pointing out this issue. In response to the poor description of the untargeted metabolomics method you mentioned, we think that it has been described in detail in the previously published article methods and supplementary materials, so this part of the details is not shown in the manuscript.
 - e. Thank you for pointing out this problem in manuscript. We have made the modification in line 509-521, 540-547 and 554-561.
 - f. We thank the reviewer for pointing out this issue. We have added this part of the data in the attached material in Table S3: Results of multiplex analysis of species and gene functions at different levels.
 - g. Thank you for pointing out this problem in manuscript. We have added the metabolite sample clustering method in the section on MATERIALS AND METHODS in line 515-521.
 - h. More descriptions have been included in the MATERIALS AND METHODS.
6. Line 181: "difference...is more significant" is fundamentally incorrect.

RESPONSE: Thank you for your professional advice, this observation is correct. We have removed the error to avoid misrepresentation.

7. Line 220: what does levels 1 and 2 mean?

RESPONSE: Thanks for your professional questions. In the KEGG PATHWAY database, biological metabolic pathways are classified into six categories: Cellular Processes, Environmental Information Processing, Genetic Information Processing, Human Diseases, Metabolism, and Organismal Systems, each of which is systematically classified into two, three, and four tiers. The second layer currently includes 43 seed pathways; the third layer is the metabolic pathway map; and the fourth layer is the specific annotation information of each metabolic pathway map. Therefore,

our expressions levels 1 and 2 refer to the first two levels of classification, respectively.

8. Please clarify: In the methods section for extraction and sequencing of metagenomic DNA, "fecal pellets from all mice were collected at the end of the acclimatization week." In the methods section for animals and treatment regime, "...mice were randomly divided into three groups after one week of acclimation". How can the metagenomic data reflect differences of the treatments if the data was acquired prior to the treatment?

RESPONSE: Thank you so much for your careful check. Experimental mice were grouped for experiments after a week of acclimatization. The intervention period was 14 weeks and fecal samples were collected at the beginning of the last week, blood was removed from the eyes at the end of the intervention, necks were broken and executed, and samples of intestine, liver and fat were collected for subsequent testing. We have revised the text to address your concerns and hope that it is now clearer in line 525.

9. The supplementary material is supposed to support the main manuscript but in its current form cannot be understood by readers.
 - a. All supplementary figure legends lack sufficient description to inform readers what they are looking at.
 - b. Supplementary figure 2 is too small to be legible.
 - c. Supplementary figure 3 by itself is meaningless. A better way is to incorporate the information into supplementary table 2 instead.
 - d. Supplementary figure 4 clearly shows pathways but the legend says "analysis of metabolites". More details have to be provided how the data presented is achieved.

RESPONSE: Thanks for your professional comments and suggestions.

- a. Based on your suggestions, we have adapted and clarified the supplementary material to more clearly express our results.
- b. Based on your suggestion, we have replaced the original image with a higher definition for identification purposes.
- c. Thank you for your suggestion, we believe that Supplementary Figure 4 (formerly

Supplementary Figure 3) corresponds to Supplementary Table 5 (formerly Supplementary Table 3) one-to-one and can more clearly represent the enrichment of different metabolic pathways, so we believe that this supplementary figure is better retained.

- d. Thank you for your suggestion, which we have described in the MATERIALS AND METHODS. functional meta-analysis pathway annotation was performed via the online website (<https://www.metaboanalyst.ca/>). This module aims to identify robust functional profiles across multiple global metabolomics datasets via two approaches: 1) integrating functional profiles from independent studies conducted under compatible LC-MS conditions; or 2) pooling peaks from complementary instruments within the same studies.
10. Although this is a follow-up study with highly similar approaches to what was published, the previous study (PMID: 32334588) was only referenced in this manuscript to escape the need to fully elaborate scientific methods. There is hardly any comparison of results of the two studies and a discussion of the comparison. What did we learn from long term probiotic intervention using engineered bacteria based that is new compared to the short-term study?

RESPONSE: Thanks for your professional comments and questions. Compared with BsS-RS06550 and natural *B. subtilis*, BsS-RS06551 has the advantage of higher butyric acid production. Moreover, the safety and effectiveness of the engineered strain was better than the original strain in short-term treatment, which has been demonstrated in the previous article. We did not compare the two studies because the experimental setups on engineered bacteria treatment and obesity are still different, and in the short-term study we did not focus on the effects of changes in the flora, but only analyzed changes at the metabolite level. This issue that you raised is also of concern to us. There are relatively few previous studies on the effect of butyric acid producing engineered bacteria on diet-induced obesity in mice, and we were not able to determine the mechanism of action of engineered bacteria in the preliminary experiments, and related mechanistic studies are in progress, and short and long term studies will be presented in future results at the same time and space level.

11. It will be useful and interesting to reanalyze the metagenomics data to determine if the different treatments affect the diversity of the gut microbial communities.

RESPONSE: Thanks for your professional comments and suggestions. Regarding the influence of

engineered bacteria on the diversity of the intestinal microbial community we show in the supplemental material (FIG S2). The results show that the effect of engineered bacteria on community richness and diversity differs across species levels.

12. It will be great if this study may be placed in the context of what is done by the community. How do the metabolic and microbial compositional changes in the gut in this study different or similar to studies involving butyrate supplementation or other high butyrate producing microbes?

RESPONSE: Thanks for your professional comments and questions. The point you made is very important to us. We did not focus on the correlation of metabolic and microbial composition changes in the gut with other high-butyrate-producing microorganisms in this study. We believe that this requires further experiments to explore the association, which needs to be demonstrated in conjunction with literature and data research. However, with our current literature research, it appears that butyrate supplementation or other high butyrate-producing microorganisms have focused more on studies of inflammatory and oncological mechanisms and relatively little attention has been paid to metabolic diseases, such as obesity.

13. The discussion section needs to undergo major correction since large sections of it should be in the materials and methods or results sections. Better organization of information within the manuscript will help improve its logic flow.

RESPONSE: Thanks for your professional comments and questions. Thank you very much for your professional and discreet comments and suggestions, we couldn't agree with you more. Based on your suggestions, we have reorganized the content of the manuscript and rewritten this part of the chapter.

Minor Comments

1. Please correct the following:
 - a. Line 77, "transparent genetic background"
 - b. "engineering bacteria" was used incorrectly as a noun multiple times in the manuscript.
 - c. Line 354: "Deferribactere"

- d. Line 519. "Turkey' test"
- e. Line 422 "Ampicillin (100 mg/L) and kanamycin (50 mg/L)" - please check your concentrations.

RESPONSE: Thank you very much for your help to improve our work. Based on your comments and suggestions, we have responded in the above content.

- a. Based on your comments, we have replaced the expression here “clear genetic background”.
 - b. Thank you for your comments, we apologize for the shoddy grammar and have replaced this misrepresentation throughout the manuscript.
 - c. Thank you for your comments, we apologize for the careless expression and have replaced this misrepresentation in the full manuscript.
 - d. Thank you for your comments, we apologize for the unprofessional expression and have replaced this misrepresentation in the manuscript.
 - e. Thank you for your comments, we have checked and confirmed the concentration of antibiotics.
2. Please provide references for the "colonization potential of bacteria".

RESPONSE: Thanks to your professional opinion, we have added 2 references to the manuscript to show that *Bacillus subtilis* has some potential for colonization (doi: 10.1038/s41586-018-0616-y, 10.1038/s41598-021-00699-4).

Reviewer: 2

Thanks to your professional opinion. We sincerely hope that this revised manuscript has addressed all your comments and suggestions.

Once again, thank you to the Editor and Reviewers for their thoughtful, diligent, and respectful critique of our paper. We’re expecting your kind understanding and hope that this revised version meets your expectations, and we are looking forward to your response.

February 16, 2022

Prof. He Huang
Tianjin University
Tianjin
China

Re: Spectrum01147-21R2 (Positive Interventional Effect of Engineered Butyrate-Producing Bacteria on Metabolic Disorders and Intestinal Flora Disruption in Obese Mice)

Dear Prof. He Huang:

Thank you for submitting your manuscript to Microbiology Spectrum. As you will see your paper is very close to acceptance. Please modify the manuscript along the lines I have recommended. As these revisions are quite minor, I expect that you should be able to turn in the revised paper in less than 30 days, if not sooner. If your manuscript was reviewed, you will find the reviewers' comments below.

When submitting the revised version of your paper, please provide (1) point-by-point responses to the issues I raised in your cover letter, and (2) a PDF file that indicates the changes from the original submission (by highlighting or underlining the changes) as file type "Marked Up Manuscript - For Review Only". Please use this link to submit your revised manuscript. Detailed instructions on submitting your revised paper are below.

Link Not Available

Sincerely,

Cheng-Yuan Kao

Reviewer comments:

Reviewer #1 (Comments for the Author):

The authors have addressed my concerns.

Additional minor comments:

Line 208: "Genetically modified" instead of "genomically modified"

Reviewer #2 (Comments for the Author):

I suggest the authors to provide validated LC-MS/MS results, including (1) retention time and (2) MS/MS, using authentic standard compounds, especially for those have been annotated in statistical analysis and metabolic pathway analysis (Fig. 6 to Fig. 8). But if there are other issues, such as insufficient time or funding, so that the validation of compound IDs can not be made, at least the author should mentioned in the manuscript that "all the compound IDs are putative". This will allow the other experts to judge this work more objectively.

Preparing Revision Guidelines

- point-by-point responses to the issues I raised in your cover letter
- Upload a compare copy of the manuscript (without figures) as a "Marked-Up Manuscript" file.
- Each figure must be uploaded as a separate file, and any multipanel figures must be assembled into one file.
- Manuscript: A .DOC version of the revised manuscript
- Figures: Editable, high-resolution, individual figure files are required at revision, TIFF or EPS files are preferred

Please return the manuscript within 60 days; if you cannot complete the modification within this time period, please contact me. If you do not wish to modify the manuscript and prefer to submit it to another journal, please notify me of your decision immediately so that the manuscript may be formally withdrawn from consideration by Microbiology Spectrum.

We appreciate the time and effort the Editor and the Reviewers dedicated to providing feedback on our manuscript and are grateful for the insightful comments and valuable improvements to our work. We have tried our best to revise the manuscript according to the construction comments and suggestions by the Reviewers. Those changes are highlighted in the manuscript. We sincerely hope that the Editor can reconsider our revised manuscript for the journal. Our response to the reviewers' comments is as follows:

Reviewer: 1

Thank you very much for your help to improve our work. Based on your suggestions, we have made the modification in the manuscript.

Reviewer: 2

Thanks for your professional opinion. Based on your comments and suggestions, we have included information on mass to charge ratio (m/z), retention time (rt) and peak area (intensity) of the identified metabolites in our previous supplemental Table S4. In response to your concerns, we have added the identification profiles of some compounds in the supplement for reference. We sincerely hope that this revised manuscript has addressed all your comments and suggestions.

Once again, thank you to the Editor and Reviewers for their thoughtful, diligent, and respectful critique of our paper. We're expecting your kind understanding and hope that this revised version meets your expectations, and we are looking forward to your response.

February 17, 2022

Prof. He Huang
Tianjin University
Tianjin
China

Re: Spectrum01147-21R3 (Positive Interventional Effect of Engineered Butyrate-Producing Bacteria on Metabolic Disorders and Intestinal Flora Disruption in Obese Mice)

Dear Prof. He Huang:

Your manuscript has been accepted, and I am forwarding it to the ASM Journals Department for publication. You will be notified when your proofs are ready to be viewed.

Sincerely,

Cheng-Yuan Kao
Editor, Microbiology Spectrum

Journals Department
Table-S6: Accept
Table-S7: Accept
Table-S5: Accept
Supplementary Material: Accept
Table-S3: Accept
Table-S4: Accept